# Inactivation of GH3.5 by COP1-mediated K63-linked ubiquitination promotes seedling hypocotyl elongation

Yongting Liu[1,2], Yinpeng Xie [2,3], Dongqing Xu [4], Xing Wang Deng [1,2,5] ✉ & Jian Li [6] ✉

CONSTITUTIVELY PHOTOMORPHOGENIC 1 (COP1), which was first discovered as a central repressor of photomorphogenesis in *Arabidopsis*, destabilizes proteins by ubiquitination in both plants and animals. However, it is unclear whether and how *Arabidopsis* COP1 mediates non-proteolytic ubiquitination to regulate photomorphogenesis. Here, we show that COP1-mediated lysine 63 (K63)-linked polyubiquitination inhibits the enzyme activity of GRETCHEN HAGEN 3.5 (GH3.5), a synthetase that conjugates amino acids to indole-3-acetic acid (IAA), thereby promoting hypocotyl elongation in the dark. We show that COP1 physically interacts with and genetically acts through GH3.5 to promote hypocotyl elongation. COP1 does not affect GH3.5 protein stability; however, it suppresses GH3.5 activity through K63-linked ubiquitination in the dark, inhibiting the endogenous conversion of IAA to IAA-amino acid conjugates. Further, light regulates IAA metabolism by suppressing the inhibitory effect of COP1 on the function of GH3.5 and its homologs. Our results shed light on the non-proteolytic role of COP1-mediated ubiquitination and the mechanism by which light regulates auxin metabolism to modulate hypocotyl elongation.

One of the most important environmental cues, light plays a crucial role in seedling morphogenesis. Hypocotyl elongation is among the most remarkable seedling morphogenic events. Rapid hypocotyl elongation in the dark (e.g., under soil) favors seedling exposure to light, and light inhibits excessive hypocotyl elongation[1]. In *Arabidopsis*, different photoreceptors have evolved to perceive specific wavelengths of sunlight. Light-activated photoreceptors suppress the accumulation and activity of central repressors to release the function of light-responsive transcription factors, thereby suppressing hypocotyl elongation and promoting photomorphogenesis[2].

The ubiquitin-proteasome system for protein degradation is conserved in eukaryotes and modulates plant growth and development processes, including seedling photomorphogenesis[3]. Ubiquitin ligase (E3) attaches ubiquitin moieties to substrates and targets them to the 26S proteasome for degradation. During light-controlled morphogenesis in *Arabidopsis* seedlings, COP1 functions as an E3 ligase and plays a critical role in destabilizing key signaling factors to repress photomorphogenesis in the dark[2]. In *Arabidopsis*, COP1 is a Really Interesting New Gene (RING)-type E3 that contains three distinct domains from its N-terminus to the C-terminus: RING, coiled-coil, and

[1]State Key Laboratory of Wheat Improvement, Peking University Institute of Advanced Agricultural Sciences, Shandong Laboratory of Advanced Agricultural Sciences in Weifang, Weifang, Shandong, China. [2]Shenzhen Key Laboratory of Plant Genetic Engineering and Molecular Design, Institute of Plant and Food Science, Department of Biology, School of Life Sciences, Southern University of Science and Technology, Shenzhen, China. [3]State Key Laboratory of Crop Stress Biology for Arid Areas/Shaanxi Key Laboratory of Apple, College of Horticulture, Northwest A&F University, Yangling, Shaanxi, China. [4]State Key Laboratory of Crop Genetics & Germplasm Enhancement and Utilization, National Center for Soybean Improvement, College of Agriculture, Nanjing Agricultural University, Nanjing, China. [5]Peking-Tsinghua Center for Life Sciences, School of Advanced Agricultural Sciences, Peking University, Beijing, China. [6]College of Life Sciences, Nanjing Normal University, Nanjing, China. ✉e-mail: deng@pku.edu.cn; jian@njnu.edu.cn

WD40 repeat domains. The RING domain normally mediates the interaction of COP1 with the ubiquitin-conjugating enzyme (E2). The coiled-coil domain mediates the interaction with accessory proteins, and the WD40 domain mediates the interaction with substrate proteins[4]. Light signaling tightly controls the E3 activity of COP1. Light-activated photoreceptors, such as phytochromes and cryptochromes, modulate the physical interaction of COP1 and its accessory proteins SUPPRESSOR OF *phyA-105* (SPAs), thereby inhibiting the E3 activity of COP1 in vivo[5,6]. Additionally, photoactivated cryptochromes prevent the interaction of COP1 with its substrate, thereby inhibiting COP1-mediated ubiquitination[7].

Ubiquitination involves successive enzymatic reactions mediated by three enzymes, namely, the ubiquitin-activating enzyme (E1), E2, and E3. E1 catalyzes the formation of a high-energy thioester bond between itself and ubiquitin, thereby activating ubiquitin. The activated ubiquitin is then transferred to E2, and E3 interacts with both E2 and the substrate and serves as a scaffold to transfer ubiquitin from E2 to the substrate. An isopeptide bond is formed between the C-terminal glycine residue of the ubiquitin and the ε-amino group of the lysine residue of the substrate[8]. A ubiquitin protein contains seven lysine (K) residues, namely, K6, K11, K27, K29, K33, K48, and K63. Each lysine on ubiquitin serves as an acceptor site for the next ubiquitin, thereby forming polyubiquitin chains with different linkages[8]. K48-linked ubiquitination is the most common ubiquitin modification type in animals and plants[9–13]. Proteins labeled with K48-linked ubiquitin chains are usually targeted to the 26S proteasome for degradation[14]. K63 ubiquitination is the second most common type. Rather than triggering protein degradation, K63 ubiquitination in animals plays a non-proteolytic role in many cases by regulating protein interaction, sub-cellular localization, and biochemical activity[15]. In plants, the function of K63 ubiquitination is involved in processes such as regulating cargo endocytosis and stabilizing proteins[16,17]. However, only a few exact E3-substrate pairs of K63 ubiquitination have been reported in plants[18–21]. Although there are more than 1400 E3s in *Arabidopsis*, only two, PLANT U-BOX 25/26 (PUB25/PUB26)[19] and ARABIDOPSIS TOXICOS EN LEVADURA 31 (ATL31)[20], have been identified as attaching K63 chains to known substrates in vivo. More than 50 substrates of *Arabidopsis* COP1 have been identified to date, all of which are degraded by COP1-mediated ubiquitination[22]. However, whether COP1 also has a non-proteolytic ubiquitination role is currently unknown.

As the first discovered phytohormone, auxin plays a fundamental role in cell expansion[23]. Inhibition of auxin biosynthesis impairs hypocotyl elongation in the dark, suggesting that auxin promotes this process[24]. Moderate auxin levels can promote cell elongation, but excessive auxin levels inhibit it[25]. Therefore, auxin homeostasis, which is regulated by auxin biosynthesis and metabolism, is crucial for auxin-driven plant growth. IAA is the predominant natural auxin in plants[26], and its inactivation of IAA mainly involves GRETCHEN HAGEN 3 (GH3) and DIOXYGENASE FOR AUXIN OXIDATION (DAO)[27]. *GH3* genes encode acyl amido synthetases that conjugate different substrates with amino acids. In *Arabidopsis*, there are 19 members of the *GH3* gene family, which are divided into three groups according to sequence homology and substrate preference[28,29]. Group II GH3 proteins, which consist of eight members (GH3.1–GH3.6, GH3.9, and GH3.17), inactivate IAA mainly redundantly by IAA conjugation with aspartic acid (Asp) or glutamic acid (Glu)[30]. The IAA-amino acid conjugates can then be irreversibly oxidized to oxIAA-amino acid conjugates by DAO proteins[31]. Genetic evidence has shown that the group II *GH3* family is redundantly involved in inhibiting hypocotyl elongation. Transgenic plants overexpressing any single group II *GH3* family member, such as *GH3.2*, *GH3.5*, and *GH3.6*, show short hypocotyls and a dwarf phenotype in the light[32–35]. In contrast, the absence of the entire group II *GH3* family in *Arabidopsis* dramatically promotes hypocotyl elongation in light[36,37]. Interestingly, auxin induces the expression of several group II *GH3s*, including *GH3.1–GH3.6*, although group II GH3s play a role in

auxin inactivation, suggesting that they act as negative feedback regulators to prevent active auxin overaccumulation[38].

Although numerous studies have uncovered the mechanisms underlying light-regulated auxin biosynthesis and signal transduction[39], it is unclear whether and how light regulates auxin metabolism. Here, we showed that COP1 attached K63 ubiquitin chains to GH3.5 in the dark and inhibited the enzymatic activity of GH3.5, rather than regulating its protein stability, thereby regulating auxin metabolism in vivo. Our results shed light on the non-proteolytic role of COP1-mediated K63 ubiquitination and provide insight into light-regulated auxin metabolism.

## Results

### Screening of potential substrates for COP1-mediated K63 ubiquitination

We used linkage-nonspecific and -specific ubiquitin antibodies to determine the global ubiquitination level in *cop1* mutants grown in the dark in order to determine whether COP1 contributes to the formation of K63 polyubiquitin chains in *Arabidopsis*. The total ubiquitination level, which represented a mixture of various ubiquitin chains, decreased significantly in the *cop1-4* and *cop1-6* mutants (Supplementary Fig. 1a). Further, similar to the double mutant of *UBC35* and *UBC36*[40,41], the E2s that specifically catalyze the formation of K63-linked ubiquitin chains in *Arabidopsis*, K63-linked ubiquitin chain accumulation was also significantly impaired in *cop1-4* and *cop1-6* (Supplementary Fig. 1b). These results suggest that COP1 is involved in the formation of K63 ubiquitin chains in the dark. Thus, we performed two types of immunoprecipitation-mass spectrometry (IP-MS) to screen potential substrates for COP1-mediated K63 ubiquitination (Supplementary Fig. 2). We used the K63 tandem ubiquitin binding entity (K63 TUBE)[42] to capture proteins with K63 ubiquitin chains from Col and *cop1-4*, and then identified proteins whose K63 ubiquitination was dependent on COP1. We used GFP-Trap® beads (ChromoTek) to co-immunoprecipitate COP1-interacting proteins from *YFP-COP1* transgenic plants[43]. The proteins that were simultaneously detected by these two strategies were suspected to be the candidates of interest (Supplementary Fig. 2). Thus, we found that GH3.6, a member of the GH3 synthetases family, was a substrate candidate for COP1-mediated K63 ubiquitination.

**COP1 genetically interacts with GH3.5 in the dark**. It has already been reported that the *GH3* synthetase family in *Arabidopsis* is divided into three groups[28,29]. Group II *GH3* genes in *Arabidopsis* consist of eight members, including *GH3.6* and its closest homolog, *GH3.5*[30]. Therefore, we investigated the genetic interaction of *COP1* and these two *GH3* genes by crossing the *gh3* mutations into *cop1-4*. As shown in Fig. 1a, b, the hypocotyls of *cop1-4 gh3.6* in the dark were the same length as those of *cop1-4*. However, in the dark, the hypocotyls of the *cop1-4 gh3.5-1* double mutants and *cop1-4 gh3.5-1 gh3.6* triple mutants were the same length and significantly longer than those of *cop1-4*, suggesting that functional *GH3.5*, but not *GH3.6*, contributes to the short hypocotyl of *cop1-4* in the dark. Knockout of four members, namely, *GH3.5*, *GH3.6*, *GH3.9*, and *GH3.17*, of group II *GH3* genes in *cop1-4* further promoted the hypocotyl elongation of *cop1-4* in the dark, indicating the obvious contribution of group II *GH3* genes to short hypocotyls in *cop1-4*. We applied nalacin, a chemical inhibitor of group II GH3 enzymes[44], to Col and *cop1-4* seedlings in the dark to further identify the effect of the *GH3* family on short hypocotyls in *cop1-4* (Fig. 1c, d). The hypocotyls of *cop1-4* were significantly elongated when treated with nalacin in the dark. These results confirm that *COP1* antagonizes the function of *GH3.5* and other group II *GH3s* to promote hypocotyl elongation in the dark.

**COP1 physically interacts with GH3.5**. As GH3.5, but not GH3.6, played a significant role in COP1-regulated hypocotyl elongation in the

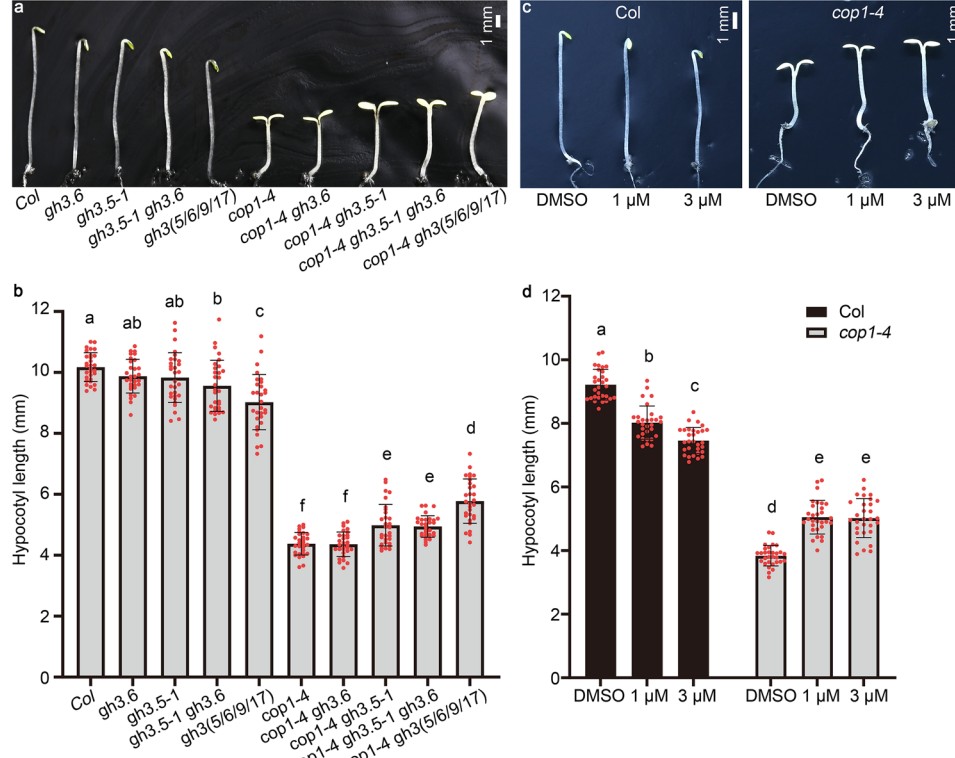

**Fig. 1 | COP1 genetically interacts with group II GH3s in the dark. a, b** Hypocotyl lengths of 4-day-old seedlings of Col, *cop1-4*, and *gh3* mutants in Col and *cop1-4* backgrounds in the dark. *gh3(5/6/9/17)*, quadruple mutant lacking *GH3.5*, *GH3.6*, *GH3.9*, and *GH3.17*. Scale bar, 1 mm. Data are presented as the mean ± standard deviation (SD), *n* = 30 seedlings. Different lowercase letters above the histogram represent statistically significant differences (*P* < 0.05) determined by ordinary one-way ANOVA with Tukey's post hoc test. **c, d** Hypocotyl lengths of Col and *cop1-*

*4* seedlings treated with different concentrations of nalacin, an inhibitor of group II GH3s. DMSO, the solvent of nalacin. Scale bar, 1 mm. Data are presented as the mean ± SD, *n* = 30 seedlings. Different lowercase letters above the histogram represent statistically significant differences (*P* < 0.05) determined by ordinary one-way ANOVA with Tukey's post hoc analysis. Source data are provided as a Source Data file.

dark (Fig. 1a, b), we investigated the physical interaction of COP1 and GH3.5. We purified recombinant GST-GH3.5 and MBP-COP1 using a prokaryotic expression system. The in vitro pull-down assay showed that COP1 interacted directly with GH3.5 (Fig. 2a). The in vivo interaction of COP1 and GH3.5 was confirmed with firefly luciferase complementation imaging (LCI) and coimmunoprecipitation (Co-IP) assays (Fig. 2b, c). Additionally, the bimolecular fluorescence complementation (BiFC) assay in *Arabidopsis* protoplasts showed a clear interaction between COP1 and GH3.5 in the nucleus (Fig. 2d). To map the subdomain of COP1 that interacted with GH3.5, we first divided full-length COP1 into the N- and C-terminal domains and found that the C-terminal domain of COP1 directly interacted with GH3.5 in vitro (Fig. 2e, f). We then examined the in vivo interaction between the truncated COP1 proteins and GH3.5 using the LCI assay in tobacco leaves. The C-terminal end of COP1 also interacted with GH3.5 in vivo (Fig. 2e, g). However, deletion of the RING, coiled-coil, or WD40 domain in COP1 alone had no effect on the interaction between COP1 and GH3.5 in vivo (Fig. 2e, g). Overall, these results demonstrate that COP1 physically interacts with GH3.5 in the nucleus and that the C-terminus of COP1 is essential for this interaction.

**COP1 has no effect on the protein stability of GH3.5 in the dark.** Because COP1 and GH3.5 antagonistically regulate hypocotyl elongation in the dark (Fig. 1), it is possible that COP1 regulates the mRNA or protein levels of GH3.5. As shown in Fig. 3a, the mRNA level of *GH3.5* was significantly decreased in *cop1-4* compared with Col in the dark, suggesting that COP1 maintains *GH3.5* expression in the dark. Prior studies have shown that *GH3.5* expression is induced by auxin, which may serve as a negative feedback mechanism to balance the auxin

pathway with growth in planta[38]. *GH3.5* expression in *cop1-4* was partially restored by the application of 2,4-dichlorophenoxyacetic acid (2,4-D), a synthetic auxin analogue (Supplementary Fig. 3), suggesting that the low *GH3.5* expression in *cop1-4* is partially due to an abnormal auxin pathway in *cop1-4*.

COP1 functions as a well-known ubiquitin ligase that promotes protein degradation. We generated a GH3.5-GFP/Col overexpressing plant, driven by the *UBQ10* promoter and introduced into *cop1-4*, in order to determine the effect of COP1 on the protein stability of GH3.5. In the dark, the abundance of GH3.5-GFP in Col was the same as that in *cop1-4* (Fig. 3b). When etiolated Col and *cop1-4* seedlings were treated with cycloheximide (CHX), an inhibitor of protein biosynthesis, the degradation rate of pre-existing GH3.5-GFP in Col was comparable to that in *cop1-4* (Fig. 3c, d). These results show that COP1 has no effect on the protein stability of GH3.5 in the dark.

**COP1 mediates K63 ubiquitination of GH3.5 in the dark.** We first performed an in vitro ubiquitination assay and found that COP1 functioned as a ubiquitin ligase to polyubiquitinate GH3.5 in vitro (Fig. 4a). Transient expression experiments were performed in *Arabidopsis* protoplasts to further test the in vivo modification of GH3.5 by COP1-mediated ubiquitination, especially K63 ubiquitination. GH3.5-GFP was transiently co-expressed with HA- and FLAG-tagged COP1 (HF-COP1)[45] in protoplasts in the dark. GH3.5-GFP proteins were immunoprecipitated from protoplasts, and their polyubiquitination level was detected with pan-ubiquitin and K63 linkage-specific ubiquitin antibodies. Compared to the sole expression of GH3.5-GFP in protoplasts, the co-expression of GH3.5-GFP and HF-COP1 in protoplasts resulted in higher levels of pan-ubiquitination and K63 ubiquitination of GH3.5-

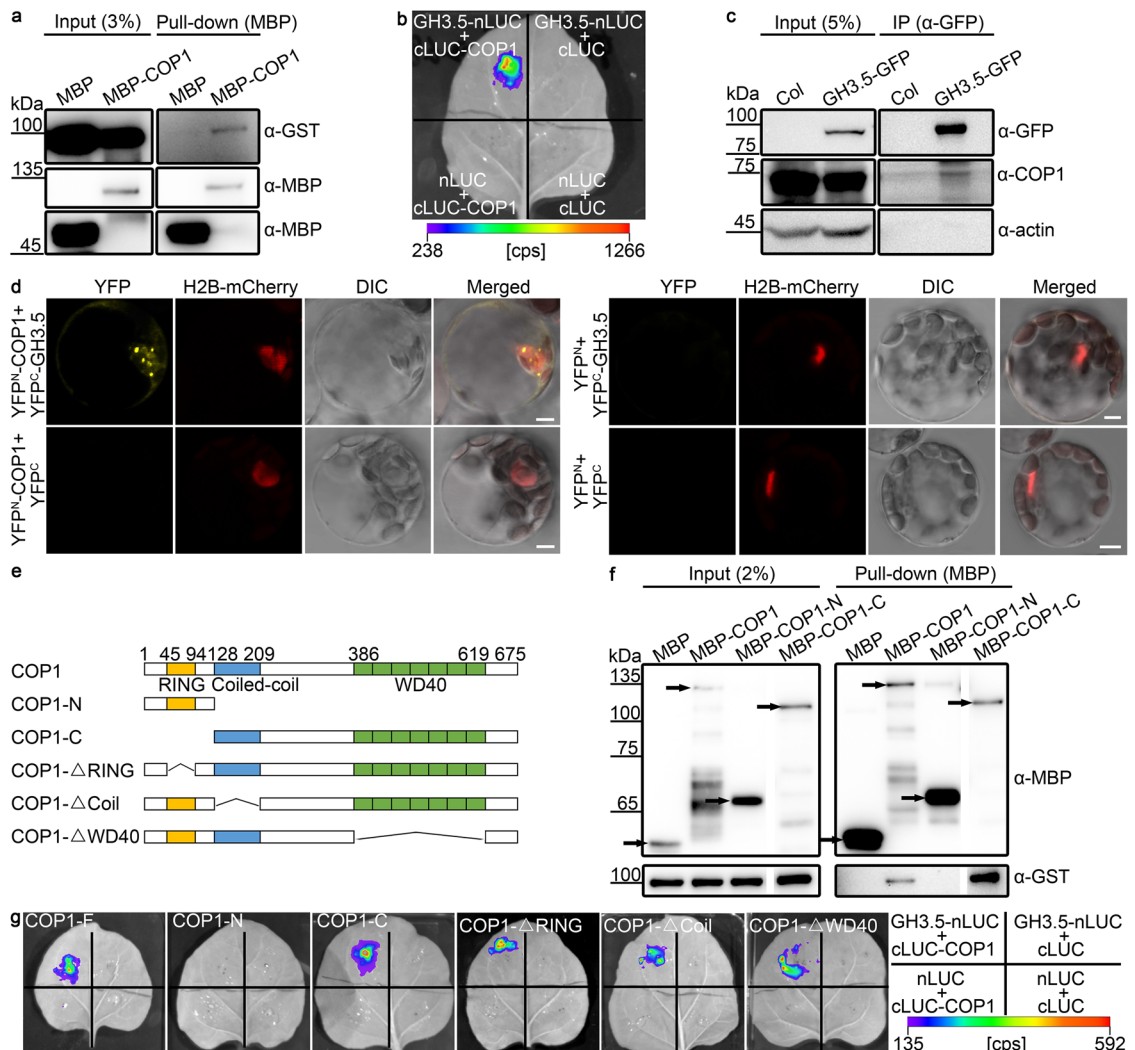

**Fig. 2 | COP1 physically interacts with GH3.5. a** In vitro pull-down assay showing the interaction between COP1 and GH3.5. COP1 was fused with MBP-tag, and GH3.5 was fused with GST-tag. α-GST, anti-GST antibody. α-MBP, anti-MBP antibody. **b** LCI assay showing the interaction between COP1 and GH3.5 in *Nicotiana benthamiana* leaves. nLUC, the vector containing the N-terminal fragment of firefly luciferase. cLUC, the vector containing the C-terminal fragment of firefly luciferase. cps, counts per second. **c** Co-IP assay showing the in vivo association of COP1 and GH3.5. *GH3.5-GFP, GH3.5-GFP* overexpression in the Col background driven by the *UBQ10* promoter. Actin was used as a loading control. **d** BiFC assay showing the interaction of COP1 and GH3.5 in *Arabidopsis* protoplasts. YFP^N, N-terminal fragment of YFP. YFP^C, C-terminal fragment of YFP. DIC, differential interference contrast imaging. H2B-mCherry was used as a nuclear marker. Scale bar, 5 μm. **e** Schematic

representations of full-length COP1 and various truncated forms. The numbers indicate the positions of the amino acids in the COP1 protein. COP1-ΔRING, COP1 lacking the RING domain. COP1-ΔCoil, COP1 lacking the coiled-coil domain. COP1-ΔWD40, COP1 lacking the WD40 domain. **f** In vitro pull-down assay showing the interaction between the C-terminus of COP1 and GH3.5. Full-length and truncated COP1 were fused with MBP-tag, and GH3.5 was fused with GST-tag. The arrows show the target proteins. **g** LCI assay showing the interaction between the C-terminus of COP1 and GH3.5 in tobacco leaves. Full-length and truncated COP1 were fused with cLUC. GH3.5 was fused with nLUC. Tobacco leaves were infiltrated with the indicated combinations. cps, counts per second. Source data are provided as a Source Data file.

GFP (Fig. 4b). Further, compared to the expression of wild-type (WT) ubiquitin proteins in protoplasts, K63R mutant ubiquitin expression resulted in impaired ubiquitination of GH3.5 by COP1 (Fig. 4c). These results suggest an important role for COP1 in the K63 ubiquitination of GH3.5 in vivo. Stably overexpressed GH3.5 proteins GH3.5-GFP/Col and GH3.5-GFP/*cop1-4* (Fig. 3) were used to determine COP1-mediated K63 ubiquitination of GH3.5 in planta. As shown in Fig. 4d, e, pan-ubiquitination of GH3.5-GFP was significantly decreased in *cop1-4* compared to Col. The K48 ubiquitination level of GH3.5-GFP in *cop1-4* was comparable to that in Col, which was consistent with the finding that COP1 had no effect on GH3.5 stability (Fig. 3b–d). However, K63 ubiquitination of GH3.5-GFP was significantly reduced in *cop1-4* (Fig. 4e). These experiments demonstrate that COP1 mainly attaches K63 ubiquitin chains to GH3.5 in vivo in the dark.

**COP1 inhibits GH3.5 activity by K63 ubiquitination.** The genetic interaction of *COP1* and *GH3.5* showed an inhibitory effect of *COP1* on *GH3.5* function in the dark (Fig. 1a, b). Because COP1 had no effect on the stability of GH3.5 in the dark (Fig. 3b–d), we determined whether COP1-mediated K63 ubiquitination inhibited the enzyme activity of GH3.5. Prior studies have shown that overexpressing group II *GH3* genes promote the conversion of IAA to IAA-amino acid conjugates, thereby increasing seedling resistance to the inhibitory effect of exogenous IAA on primary root growth[30,46]. The primary root growth of *cop1-4* was significantly resistant to exogenous IAA in the dark (Fig. 5a). Further, the loss of *GH3.5* in *cop1-4* restored the sensitivity of *cop1-4* to IAA treatment (Fig. 5a), indicating that the hyposensitivity of *cop1-4* to IAA may be due to the increased function of *GH3.5* in *cop1-4*. A prior study has shown that nalacin inhibits the primary root growth of

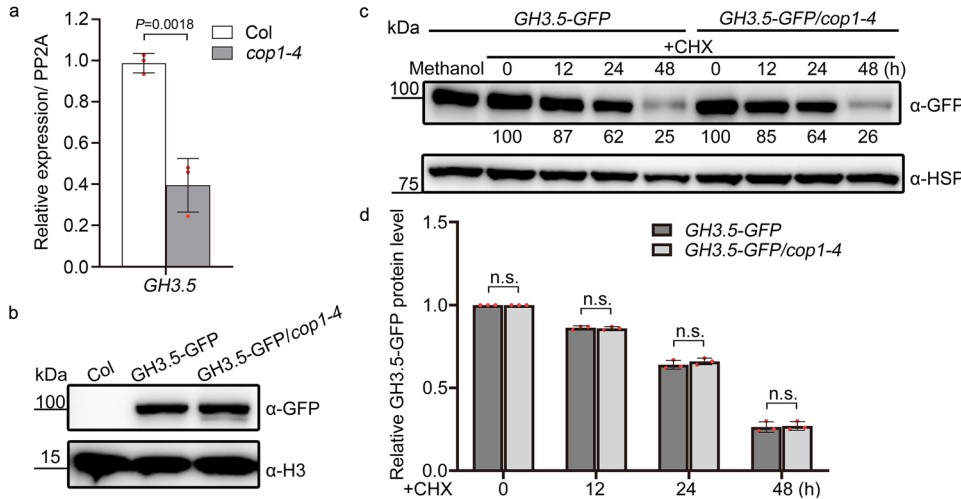

**Fig. 3 | COP1 has no effect on the protein stability of GH3.5 in the dark. a** Relative *GH3.5* expression in dark-grown Col and *cop1-4* seedlings. Relative transcript levels were normalized to *PP2A*. Data are presented as the mean ± SD, *n* = 3 technical replicates. The statistically significant difference was determined by two-sided Student's *t* test. The experiment was independently repeated twice with similar results. **b** GH3.5-GFP protein levels in dark-grown Col, *GH3.5-GFP*, and GH3.5-GFP/*cop1-4* seedlings. *GH3.5-GFP*, *GH3.5-GFP* overexpression in the Col background driven by the *UBQ10* promoter. H3 was used as a loading control. **c** Degradation rates of GH3.5-GFP proteins in Col and *cop1-4* backgrounds with CHX treatment. Four-day-old seedlings were treated with 500 μM CHX, an inhibitor of protein biosynthesis, for the indicated times in the dark. HSP was used as a loading control. The numbers below the lanes indicate the relative band intensities of GH3.5-GFP. **d** The relative quantification of GH3.5-GFP levels according to the results from (**c**). GH3.5-GFP levels were normalized to the HSP loading control. Data are presented as the mean ± SD, *n* = 3 biological replicates. n.s. not significant in two-sided Student's *t* test. Source data are provided as a Source Data file.

seedlings by inhibiting group II GH3 activity[44]. We thus examined the primary root growth of *cop1-4* when treated with nalacin in the dark. As shown in Fig. 5b, the primary root growth of *cop1-4* was hyposensitive to nalacin treatment compared to Col, indicating increased GH3 activity in *cop1-4* in the dark.

We then prepared GST-GH3.5 proteins linked to K63 ubiquitin chains (GST-GH3.5-K63) by COP1-mediated ubiquitination in vitro (Fig. 5c), and we performed an in vitro enzyme reaction to analyze the effect of COP1-mediated K63 ubiquitination on GH3.5 activity. Compared to GST-GH3.5 without ubiquitin chains (GST-GH3.5), the yield of IAA-Asp conjugated by GST-GH3.5-K63 decreased dramatically (Fig. 5d, e). Kinetic analysis of the enzymatic reaction further revealed an almost 6-fold reduction in the catalytic activity of GST-GH3.5-K63 compared to GST-GH3.5 proteins (Fig. 5f). $K_m^{IAA}$ was only slightly lower for GST-GH3.5-K63 than that for GST-GH3.5 (Fig. 5f). These results suggest that COP1-mediated K63 ubiquitination can strongly inhibit the catalytic activity of GH3.5 synthetase in vitro, possibly through noncompetitive inhibition. We measured the levels of IAA and IAA-amino acid conjugates in the etiolated seedlings to further verify GH3.5 activity inhibition by COP1 in vivo (Fig. 5g). The free IAA content in *cop1-4* was comparable to that in Col. However, the contents of IAA-amino acid conjugates, such as IAA-Asp and IAA-Glu, were significantly higher in *cop1-4* than in Col. The loss of *GH3.5* in *cop1-4* significantly reduced the IAA-Asp content in *cop1-4*. Knockout of more group II *GH3* genes (including *GH3.5*, *GH3.6*, *GH3.9*, and *GH3.17*) in *cop1-4* even more significantly reduced the IAA-Asp and IAA-Glu conjugate levels in *cop1-4*. Altogether, these results demonstrate that COP1-mediated K63 ubiquitination of GH3.5 inhibits its activity in the dark.

**Light regulates IAA metabolism by suppressing the function of COP1.** Because COP1 is a key repressor of the light signaling pathway and its function is suppressed by light via multiple regulatory mechanisms, we determined whether light also regulated GH3.5 activity in a COP1-associated manner. We first tested the possible role of light in regulating the GH3.5 mRNA and protein levels. RT-qPCR data showed that *GH3.5* expression was significantly lower in continuous white light (cWL) than in continuous darkness (cD), and it gradually

decreased when Col seedlings switched from dark to light (Supplementary Fig. 4a). Additionally, the differential expression of *GH3.5* in dark-grown Col and *cop1-4* seedlings disappeared when the seedlings were grown in the light (Supplementary Fig. 4b), suggesting that light partially inhibits *GH3.5* expression by repressing the function of *COP1*. The GH3.5-GFP/Col overexpressing line was used to analyze the protein stability of GH3.5 regulated by light. The protein levels of GH3.5 did not differ when the seedlings were grown under different light conditions (Supplementary Fig. 4c), indicating that light had no effect on the GH3.5 protein stability.

To avoid the interference caused by the light-controlled expression of *GH3.5*, we used a *GH3.5-GFP* overexpressing line (Fig. 3) to examine the light-regulated K63 ubiquitination of GH3.5. Both GH3.5-GFP/Col and GH3.5-GFP/*cop1-4* seedlings were grown in cD or cWL. GH3.5-GFP proteins were immunoprecipitated from these materials and used to analyze the K63 ubiquitination level. GH3.5-GFP/Col seedlings grown in the light showed a significantly lower K63 ubiquitination level for GH3.5-GFP compared to dark-grown seedlings (Fig. 6a), suggesting that light can suppress K63 ubiquitination of GH3.5. The K63 ubiquitination level of GH3.5-GFP in *cop1-4* did not differ in either dark or light, and it was significantly lower than the level in Col grown in the dark (Fig. 6a). This showed that inhibiting the function of COP1 is required for light-suppressed K63 ubiquitination of GH3.5.

To further confirm the in vivo enhancement of GH3.5 activity by light, we measured the IAA and IAA-Asp conjugate levels in seedlings grown in cD, as well as those grown in cWL. In Col, light had no effect on the endogenous IAA content (Fig. 6b) but suppressed IAA-Asp accumulation (Fig. 6c) and promoted oxIAA-Asp accumulation (Fig. 6d), the oxidized product of IAA-Asp (Fig. 6c). A prior study has shown that IAA-Asp can be directly converted to oxIAA-Asp by DAO1[31]. Therefore, the lower content of endogenous IAA-Asp in the light might be due to the faster conversion of IAA-Asp to oxIAA-Asp. As shown in Fig. 6e, the average content of endogenous IAA-Asp and its oxidized form (IAA-Asp + oxIAA-Asp) in light-grown Col was about 62.04 ng g⁻¹ F.W., which was much higher than in dark-grown Col (about 20.90 ng g⁻¹ F.W.). These results suggest that light promotes the

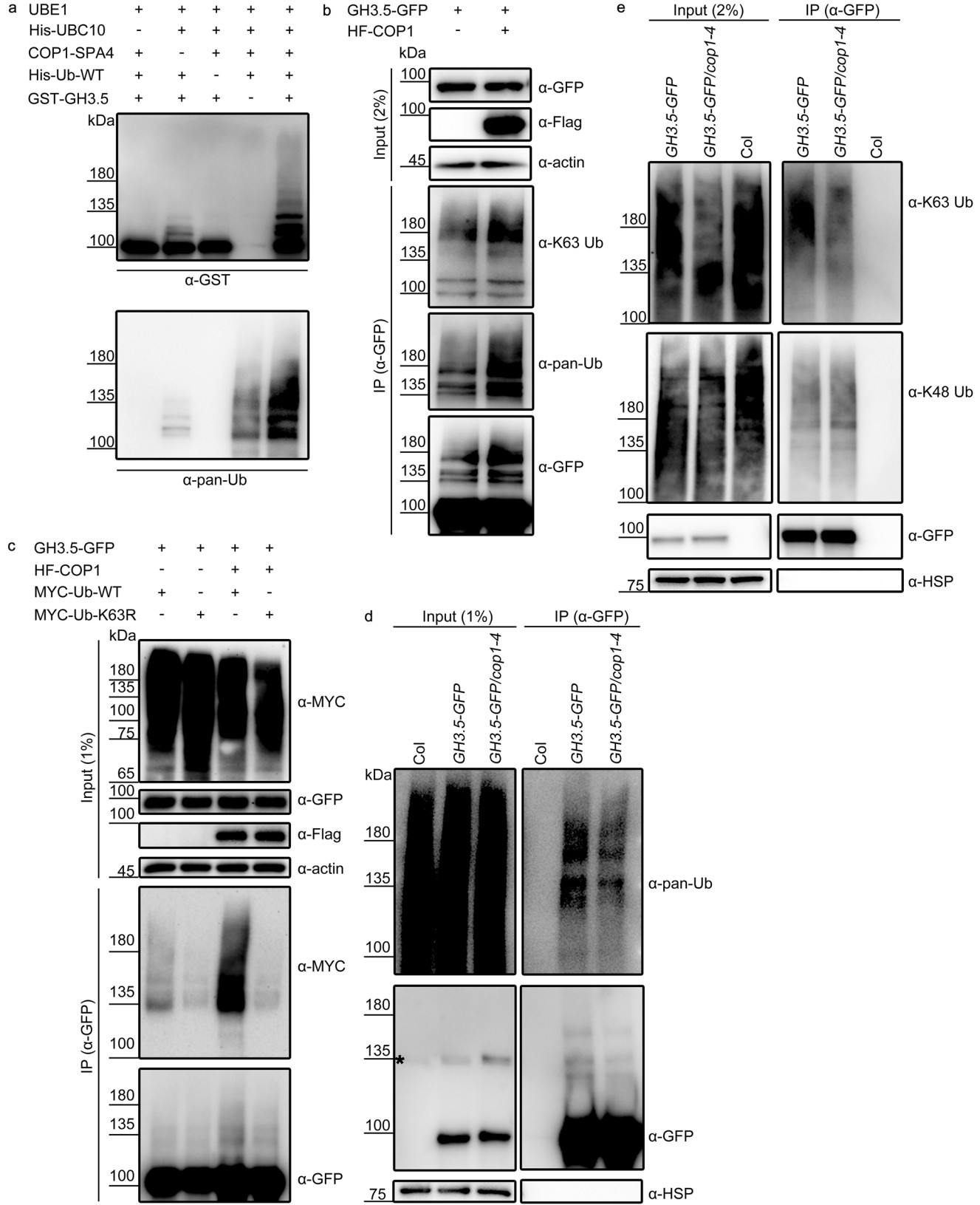

production of different IAA-Asp conjugates, consisting of IAA-Asp and oxIAA-Asp. Light-promoted production of IAA-Asp + oxIAA-Asp was significantly attenuated in the *gh3(5/6/9/17)* quadruple mutant (Fig. 6e), suggesting that light promotes IAA-Asp + oxIAA-Asp production in a *GH3*-dependent manner. Further, the loss of function of *COP1* resulted in constitutive IAA-Asp + oxIAA-Asp overaccumulation in both dark

and light (Fig. 6e), suggesting a negative role of COP1 in the light-promoted accumulation of IAA-Asp + oxIAA-Asp. Knockout of *GH3.5*, *GH3.6*, *GH3.9*, and *GH3.17* in *cop1-4* again inhibited the excessive IAA-Asp + oxIAA-Asp accumulation in *cop1-4* in the light (Fig. 6e). To further test whether light-regulated IAA metabolism contributes to the modulation of hypocotyl elongation, we analyzed the hypocotyl lengths of

**Fig. 4 | COP1 mediates K63-linked ubiquitination of GH3.5 in the dark. a** In vitro ubiquitination assay showing that COP1 ubiquitinates GH3.5. UBE1 (E1), His-UBC10 (E2), COP1-SPA4 complex (E3), and His-tagged ubiquitin were used for the assay. Ubiquitinated GST-GH3.5 was detected with anti-GST and anti-pan-Ub antibodies. **b** COP1-mediated ubiquitination of GH3.5 in *Arabidopsis* protoplasts. *GH3.5-GFP* plasmids alone or together with *HF-COP1* plasmids were each transfected into protoplasts. Immunoprecipitated GH3.5-GFP proteins were used to detect the ubiquitination levels of the proteins. Actin was used as a loading control. **c** COP1-mediated ubiquitination of GH3.5 with the co-expression of the K63R mutant ubiquitin in protoplasts. MYC-Ub-WT, MYC-tagged wild-type ubiquitin. MYC-Ub-K63R, MYC-tagged ubiquitin with Lys63 substituted by arginine (R). Ubiquitination of immunoprecipitated GH3.5-GFP proteins was detected with an anti-MYC antibody. Levels (**d**) and linkage types (**e**) of ubiquitination of GH3.5-GFP in Col, *GH3.5-GFP*, and GH3.5-GFP/*cop1-4* seedlings grown in the dark. GH3.5-GFP proteins were immunoprecipitated with GFP-trap magnetic beads, and their ubiquitination was detected with anti-pan-Ub, anti-K48 Ub and anti-K63 Ub antibodies. The asterisk in (**d**) indicates a non-specific band. HSP was used as a loading control. Source data are provided as a Source Data file.

seedlings grown in cD or cWL (Supplementary Fig. 5). Light-inhibited hypocotyl elongation was significantly attenuated in the *gh3(5/6/9/17)* mutant (Supplementary Fig. 5), which was consistent with the reduced production of IAA-Asp + oxIAA-Asp in *gh3(5/6/9/17)* when exposed to light (Fig. 6e). Moreover, knockout of the four *GH3s* in *cop1-4* partially restored hypocotyl elongation of *cop1-4* when under light (Supplementary Fig. 5), indicating the existence of an inverse correlation between hypocotyl elongation and IAA-Asp + oxIAA-Asp accumulation in *cop1-4* (Fig. 6e). These results show that light regulates IAA metabolism by suppressing the inhibitory effect of COP1 on the function of group II GH3s, thereby modulating hypocotyl elongation.

## Discussion

Numerous studies have established a comprehensive regulatory network based on COP1-mediated ubiquitination and the degradation of key signaling factors in the regulation of plant growth and development[22]. However, the non-proteolytic role of ubiquitination in plants is gradually coming into focus[19,47]. In this study, we revealed that non-proteolytic ubiquitination of GH3.5 by COP1 promotes hypocotyl elongation in the dark (Fig. 7). In the dark, functional COP1 physically interacts with GH3.5 in the nucleus and mediates the K63 ubiquitination of GH3.5. The enzyme activity of GH3.5 is inhibited by ubiquitination, limiting the conversion of IAA to IAA-Asp conjugates, thereby promoting hypocotyl elongation. In the light, COP1 inactivation impairs K63 ubiquitination of GH3.5, leading to GH3.5 activation. Activated GH3.5 efficiently promotes the conversion of IAA to IAA-Asp conjugates, thereby inhibiting hypocotyl elongation. Thus, the COP1-mediated K63 ubiquitination of GH3.5 plays an important role in light-regulated auxin metabolism and hypocotyl elongation (Fig. 7).

Our results showed an interaction between COP1 and GH3.5 in the nucleus (Fig. 2d), indicating that COP1 modulates GH3.5 activity through K63 ubiquitination in the nucleus (Fig. 7). However, the subcellular localization of group II GH3s in *Arabidopsis* remains unclear[38]. The GH3s of several other species that produce IAA-amino acid conjugates have been reported to be localized in both the cytoplasm and the nucleus[48–50]. Moreover, the GFP fluorescence signals from the GH3.5-GFP/Col line indicate that the GH3.5 proteins are localized in both the cytoplasm and nucleus (Fig. 3, Supplementary Fig. 6), supporting the nuclear interaction between COP1 and GH3.5 (Figs. 2d and 7). Classical auxin perception and signaling transduction by TRANSPORT INHIBITOR RESPONSE1/AUXIN SIGNALING F-BOX PROTEIN (TIR1/AFB) receptors occur in the nucleus, and are critical for regulating gene expression and plant growth and development[51,52]. It is likely that the nuclear localization of COP1-GH3.5 regulation provides a rapid mechanism to modulate the nuclear auxin pathway to regulate gene expression.

It has been well established that COP1 targets key signaling factors for ubiquitination and proteasomal degradation in the dark[22]. In contrast, we demonstrated the non-proteolytic role of COP1-mediated ubiquitination in this study. Prior reports have shown that the same E3 ligase can ubiquitinate different substrates with different ubiquitin chains in mammals and plants. For example, CELLULAR INHIBITOR OF APOPTOSIS-1 (cIAP1), a mammalian RING-type E3 ligase, directs proteasomal degradation of RECEPTOR-INTERACTING PROTEIN KINASE 1 (RIPK1) through K48 ubiquitination, thereby repressing cell death[53]. K63 ubiquitination of RECEPTOR INTERACTING PROTEIN 1 (RIP1), mediated by cIAPs, alters RIP1-containing signaling complex assembly and thus facilitates cancer cell survival[54]. PUB25/PUB26 target BOTRYTIS-INDUCED KINASE 1 (BIK1) for degradation to suppress the immune response of *Arabidopsis*[55]. Otherwise, they can restore MYB15 transcription factor accumulation through K63 ubiquitination in response to prolonged cold exposure[19]. IPA1 INTERACTING PROTEIN1 (IPI1) is also a RING domain-containing E3 ligase in *Oryza sativa*. It ubiquitinates the same substrate, IDEAL PLANT ARCHITECTURE1 (IPA1), with different ubiquitin chains in different tissues to modulate the architecture of rice[18]. Further efforts are required to decipher the mechanism by which COP1 determines the linkage types of ubiquitin chains. With regard to RING-type E3-mediated ubiquitination, linkage specificity is mainly determined by the interactive E2 enzyme[56,57]. UBC13, a member of the E2 family, specifically catalyzes K63 ubiquitin chain formation in yeast, mammals, and plants[16]. The RING-type E3 ligase cIAP1 can cooperate with UBC13 to catalyze K63 ubiquitination of RIP1 in mammals[54]. It is reasonable to assume that COP1 can pair up with UBC35/UBC36 (UBC13 homologs in *Arabidopsis*)[41] to mediate K63 ubiquitination of GH3.5. Additionally, COP1 conditionally serves as a subunit of a multimeric CULLIN4-DNA DAMAGE BINDING PROTEIN 1^COP1–SPA (CUL4–DDB1^COP1–SPA) E3 ligase complex. In this E3 complex, COP1-SPA functions as a substrate receptor to recruit substrate proteins, while RING BOX 1 (RBX1), an evolutionarily conserved RING domain-containing protein, recruits E2 enzymes to the complex[58,59]. The mammalian CULLIN1-based E3 ligase complex likely recruits UBC13 via RBX1 to promote K63 ubiquitination of the substrate[60]. Whether CUL4–DDB1^COP1–SPA E3 complex also recruits UBC35/UBC36 via RBX1 to mediate K63 ubiquitination in *Arabidopsis* remains to be investigated in the future. It also remains a possibility that since COP1 itself has the RING domain for potential E3 catalytic activity[61,62], it is also feasible that E3 activity for K63 ubiquitination is derived only from COP1/SPA complexes (or even free COP1 alone), without CUL4 association. In the case of legume–rhizobium symbiosis, phosphorylation of *Lotus japonicus* NFR-INTERACTING RING-TYPE E3 LIGASE 1 (NIRE1) in response to rhizobium inoculation induces the switch of NIRE1 from K48 to K63 ubiquitination activity[21]. COP1 has been reported to be modified at the post-translational level, for example by ubiquitination, SUMOylation, and phosphorylation[43,63,64]. Whether these modifications are responsible for the selection of chain linkages by COP1 should also be considered.

*GH3.5* is a member of the group II *GH3* family, which comprises eight members and redundantly catalyzes IAA to IAA-amino acid conjugates[38]. Our results showed that functional *GH3.5* and its three homologous genes contribute to short hypocotyls in *cop1-4* (Fig. 1). To further assess the role of group II *GH3s* in *COP1*-regulated hypocotyl elongation, we mutated *COP1* in the Col and *gh3-septuple* (knockout of the entire group II *GH3* family except *GH3.9*) lines. The *cop1-4* mutant has a C-to-T mutation in *COP1* genomic DNA, which changes the Gln^283 codon (CAA) to a stop codon (UAA)[65]. According to the mutational information of *cop1-4*, we generated independent *cop1-4 like* (*cop1-4l*) mutants in Col and *gh3-septuple* backgrounds using the CRISPR/Cas9 system (Supplementary Fig. 7a). The hypocotyls of the *cop1-4l*

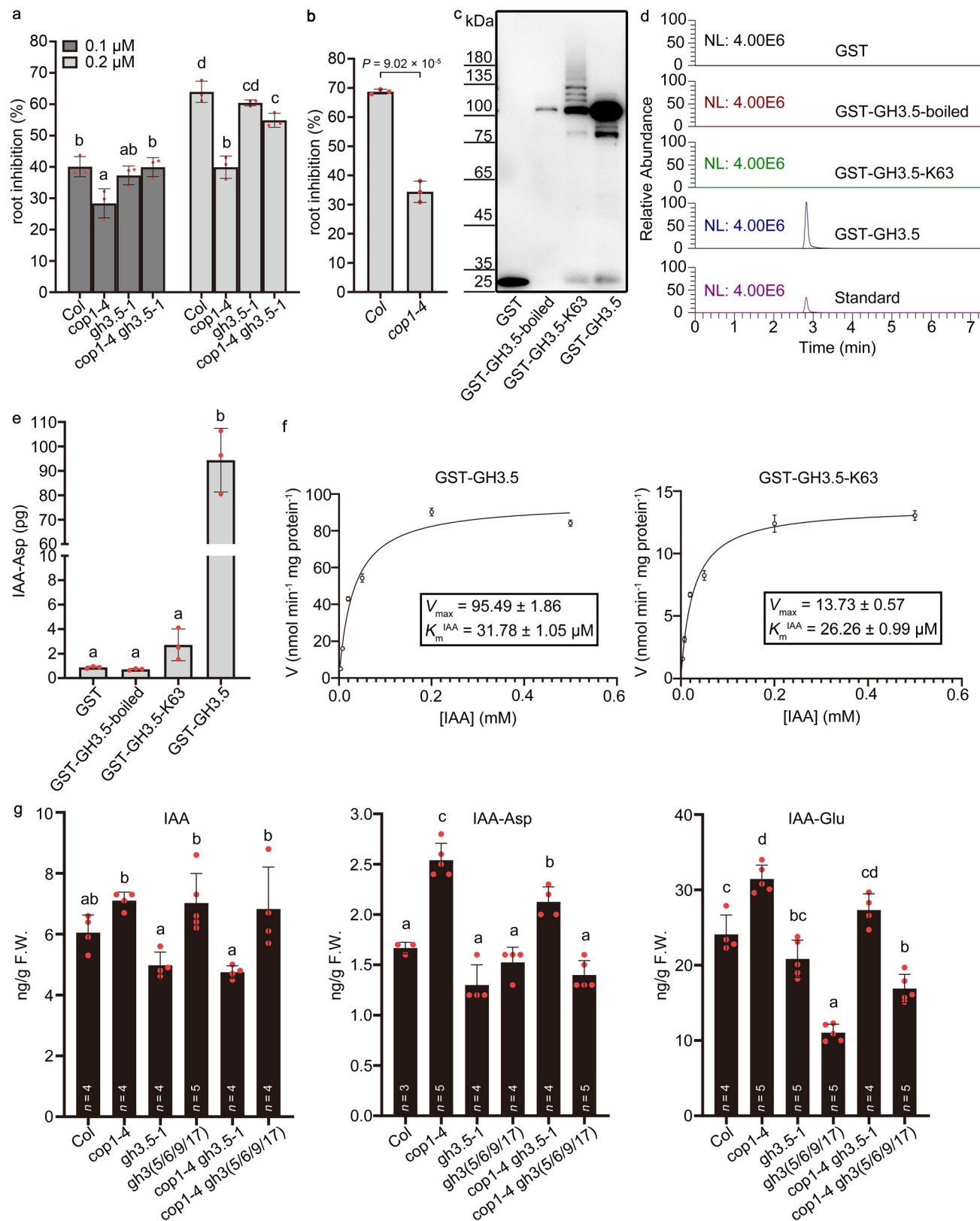

*gh3-septuple* mutant were the same length as those of the *gh3-septuple* mutant and much longer than those of the *cop1-4l* single mutant (Supplementary Fig. 7b, c). This genetic interaction indicates that the entire group II *GH3* family plays a very important role in *COP1*-regulated hypocotyl elongation. It will be worthwhile to analyze the correlation between *COP1* and the individual members of group II *GH3s* in detail in the future.

Our results showed that COP1-mediated K63 ubiquitination suppresses GH3.5 enzyme activity (Figs. 4 and 5). Nonetheless, the molecular mechanism of the inhibition of GH3.5 activity by ubiquitination is still unclear. Our kinetic analysis of the enzymatic reaction showed only a slight decrease in $K_m^{IAA}$ for GH3.5 when GH3.5 was modified by COP1-mediated K63 ubiquitination (Fig. 5f), suggesting that this ubiquitination does not affect the affinity of GH3.5 for IAA in vitro. GH3.5

**Fig. 5 | COP1-mediated K63 ubiquitination inhibits GH3.5 activity. a** Auxin inhibition of primary root growth in Col, *cop1-4*, *gh3.5-1*, and *cop1-4 gh3.5-1*. Seedlings were grown in the dark on MS medium with IAA or without IAA (0, 0.1, and 0.2 μM). IAA-mediated inhibition of root growth is expressed as the percentage decrease in the root length of IAA-treated seedlings compared to control seedlings. Different lowercase letters above the histogram represent statistically significant differences ($P < 0.05$) determined by ordinary two-way ANOVA with Tukey's post hoc test. **b** Nalacin inhibition of primary root growth in *cop1-4*. Etiolated seedlings were treated with 6 μM nalacin. Root growth inhibition is expressed as the percentage decrease in length relative to seedlings grown without nalacin. Data are presented as the mean ± SD, $n = 3$ biological replicates. For each replicate, $n = 22$ and 15 seedlings in (**a**) and (**b**), respectively. Statistically significant differences were determined by two-sided Student's *t* test. **c** Preparation of GST-GH3.5 proteins linked to K63 ubiquitin chains by COP1. GST-GH3.5-boiled, inactive GH3.5 caused by

boiling; GST-GH3.5-K63, GH3.5 linked to K63 chains by COP1; GST-GH3.5, GH3.5 without ubiquitination. The IAA-Asp produced by GH3.5 was quantified by analyzing the peak area from the UPLC-MS assay. The extracted ion chromatograms of IAA-Asp products are shown in (**d**), and the quantification of the IAA-Asp yield is shown in (**e**). NL, normalization level. Standard, authentic IAA-Asp standard. Data in (**e**) are presented as the mean ± SD ($n = 3$ biological replicates), statistically significant differences ($P < 0.001$) were determined by ordinary one-way ANOVA with Tukey's post hoc test. **f** Michaelis–Menten diagram and kinetic parameters of GH3.5 for IAA-Asp. Initial velocities are plotted against the IAA concentration. The average values ± SD ($n = 3$ biological replicates) are shown. **g** Content of endogenous IAA and its conjugates in the dark. Data are presented as the mean ± SD ($n =$ number of biological replicates). Statistically significant differences ($P < 0.05$) were determined by ordinary one-way ANOVA with Tukey's post hoc test. Source data are provided as a Source Data file.

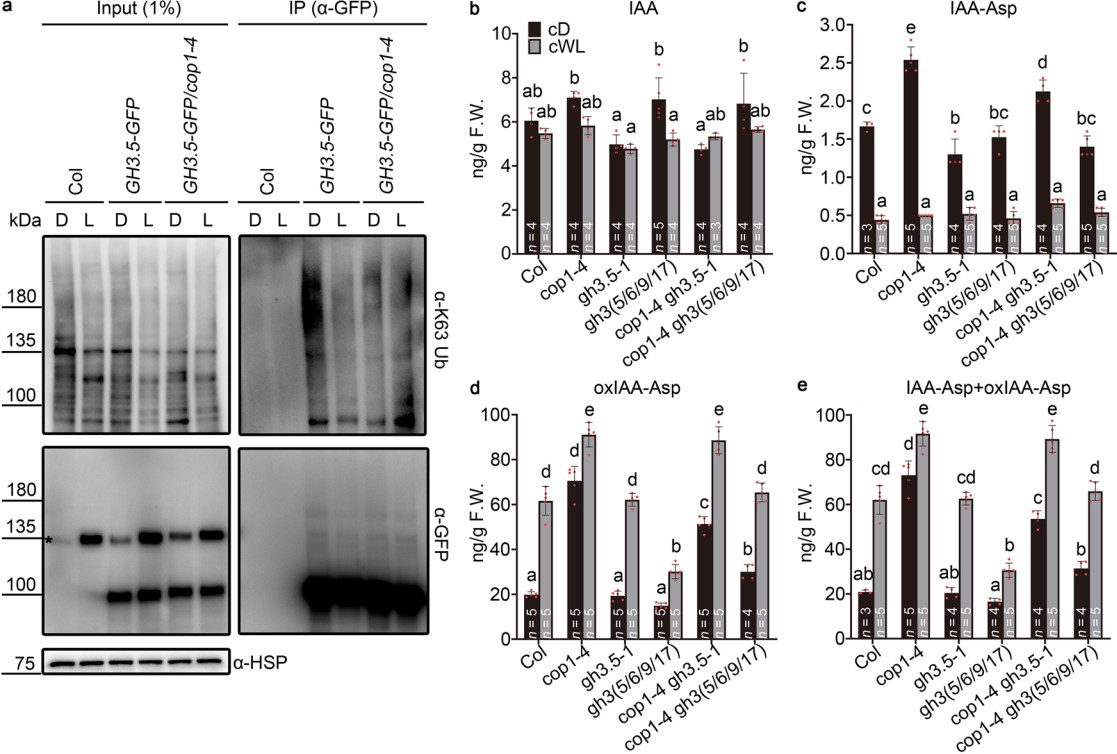

**Fig. 6 | Light regulates IAA metabolism by suppressing COP1 function. a** Light suppresses the COP1-mediated K63 ubiquitination of GH3.5. Col, *GH3.5-GFP*, and GH3.5-GFP/*cop1-4* seedlings were grown in continuous darkness or continuous white light (35 μmol m$^{-2}$ s$^{-1}$) for 4 days. Immunoprecipitated GH3.5-GFP proteins were detected with anti-K63 Ub and anti-GFP antibodies. The asterisk indicates a non-specific band. HSP was used as a loading control. D, continuous darkness; L, continuous white light. **b**–**e** Content of endogenous IAA, IAA-Asp, oxIAA-Asp, and

IAA-Asp + oxIAA-Asp in the dark or light. Whole seedlings grown for 4 days in the dark or light (35 μmol m$^{-2}$ s$^{-1}$) were used to measure IAA, IAA-Asp and oxIAA-Asp contents. cD, continuous darkness; cWL, continuous white light. Data are presented as the mean ± SD ($n =$ number of biological replicates). Different lowercase letters above the histogram represent statistically significant differences ($P < 0.05$) determined by ordinary two-way ANOVA with Tukey's post hoc test. Source data are provided as a Source Data file.

catalyzes IAA-Asp formation via a two-step reaction mechanism. In the first step, the intermediate IAA-adenosine monophosphate (IAA-AMP) is formed in the presence of adenosine triphosphate (ATP); in the second step, AMP is displaced by Asp, resulting in the end product IAA-Asp[66]. Similar to the crystal structures of other members of the GH3 family[44,67], GH3.5 has a large N-terminal domain and a small C-terminal domain, and they are connected by a flexible hinge loop[44]. The active catalytic sites are located at the interface of the N- and C-terminal domains. The rotation of the C-terminal domain through the hinge loop allows residues around the active site to interact with substrates, which is necessary for the success of a continuous two-step enzymatic reaction[44,66,67]. The binding of K63 ubiquitin chains to GH3.5 by COP1 may disrupt this conformational change, thereby inhibiting the GH3.5-

mediated enzymatic reaction. Further efforts to identify the ubiquitination sites on GH3.5 will be helpful in investigating the structural mechanism underlying the inhibition of GH3.5 activity by COP1-mediated ubiquitination.

In summary, our study reveals the non-proteolytic role of COP1-mediated K63 ubiquitination in the regulation of photomorphogenesis and sheds light on the mechanism by which light regulates auxin metabolism to modulate seedling photomorphogenesis.

## Methods

### Plant materials, growth conditions and phenotypic analysis

The *gh3.5-1* (SALK_033434C), *gh3.6* (SALK_013458C), *gh3(5/6/9/17)*, *gh3(1/2/3/4/5/6/17)* septuple mutant, *ubc35-1 ubc36-1*, *cop1-4*, *cop1-6*,

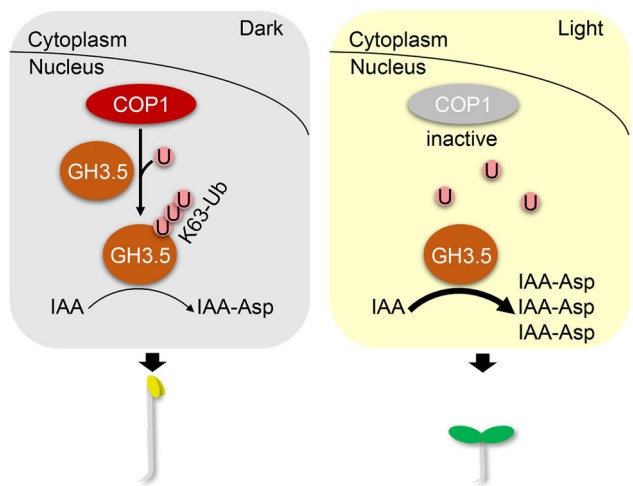

**Fig. 7 | A working model illustrating how COP1-mediated K63 ubiquitination regulates hypocotyl elongation.** U ubiquitin protein, K63-Ub K63-linked ubiquitin chain.

and *YFP-COP1/cop1-6* have been reported previously[37,40,41,43,65,68,69]. Seeds were surface sterilized with 15% (v/v) sodium hypochlorite solution for 10 min and then grown on Murashige and Skoog (MS) medium (Sigma-Aldrich, pH 5.7) supplemented with 1% (w/v) sucrose (Sangon Biotech) and 0.6% (w/v) agar (Sigma-Aldrich). After stratification at 4 °C in the dark for 2 days, the plates were transferred to continuous white light (cWL, 50 μmol m$^{-2}$ s$^{-1}$) at 22 °C for 12 h to induce seed germination. Seeds were then grown at 22 °C in continuous darkness or under other light conditions as indicated. Tobacco (*Nicotiana benthamiana*) plants were grown in a greenhouse under a 16 h light/8 h dark cycle at 28 °C. Four-day-old etiolated *Arabidopsis* seedlings were photographed with a camera (EOS80D, Canon), and hypocotyl or root lengths were calculated with ImageJ software.

**Plasmid construction and generation of transgenic plants**
To generate pSY736-*YFP$^N$-COP1* and pSY735-*YFP$^C$-GH3.5* constructs for the BiFC experiment, the coding sequence (CDS) of *COP1* was amplified by PCR and cloned into the *Sal*I/*Spe*I sites of the pSY736 vector[70], *GH3.5* CDS was amplified and inserted into the *Spe*I/*Bam*HI sites of the pSY 735 vector[70].

To generate *cLUC-COP1* constructs for LCI assays, the full-length CDS and various truncated CDS of *COP1* were cloned into the *Kpn*I/*Sal*I sites of the pCAMBIA1300-*cLUC* vector[71]. To generate the *GH3.5-nLUC* construct, the full-length CDS of *GH3.5* was inserted into the *Kpn*I/*Sal*I sites of the pCAMBIA1300-*nLUC* vector[71].

To generate pSY738-*35S:2×MYC-Ub* and pSY738-*35S:2×MYC-Ub-K63R* constructs for transient expression in *Arabidopsis* protoplasts, *2×MYC* and *ubiquitin* (*Ub*) DNA fragments were amplified individually, *Ub-K63R* mutation DNA was obtained by primer-based site-directed mutagenesis using wild-type *Ub* as template, then *2×MYC* together with *Ub/Ub-K63R* was cloned into the *Sal*I/*Not*I sites of the pSY 738 vector[70] using the ClonExpress II One Step Cloning Kit (Vazyme).

The N- and C-terminal DNA fragments of *COP1* were cloned into the *Eco*RI/*Sal*I sites of the pMAL-c2X vector, to generate the pMAL-c2X-*MBP-COP1-N* and pMAL-c2X-*MBP-COP1-C* constructs. For purification of GST-GH3.5 and His-UBC10 recombinant proteins, the *GH3.5* CDS was inserted into the *Bam*HI/*Eco*RI sites of the pGEX4T-1 vector, the *UBC10* CDS was amplified from the Col cDNA and inserted into the *Bam*HI/*Eco*RI sites of the pET28a vector. To generate *His-Ub* and *His-Ub-K63* (all lysine residues of ubiquitin were substituted by arginine, except K63) constructs, *Ub* and *Ub-K63* fragments were cloned into the *Bam*HI/*Eco*RI sites of the pET28a vector.

To generate *UBQ10$_{pro}$:GH3.5-GFP* transgenic plants, *Arabidopsis UBQ10* fragment[72], *GH3.5* and *GFP* CDS fragments were recombined into the *Sal*I/*Bam*HI sites of the pCAMBIA1300 vector using the ClonExpress II One Step Cloning Kit (Vazyme). The *Agrobacterium tumefaciens* strain GV3101 carrying the construct was used to transform Col plants. Homozygous transgenic plants were screened on MS medium with an addition of 50 mg/L hygromycin. The *cop1-4 like* (*cop1-4l*) mutants were generated using the CIRSPR/Cas9 system[73] in the background of Col and *gh3-septuple*. Two target sequences for *COP1* were selected using the CRISPR-P web tool (http://crispr.hzau.edu.cn/CRISPR/)[74]. All constructs were confirmed by DNA sequencing. The primer sequences used in this study are listed in Supplementary Table 1.

**Western blot analysis**
*Arabidopsis* seedlings were harvested, ground into powder, and lysed in protein extraction buffer containing 8 M urea, 100 mM NaH$_2$PO$_4$, 100 mM Tris-HCl (pH 8.0), 1× phenylmethanesulfonyl fluoride (PMSF) and 1× protease inhibitor cocktail (Roche). Samples were centrifuged at 13,000 × *g* for 10 min at 4 °C, then the supernatants were transferred into new tubes. The total protein concentration was quantified using the Bradford assay reagent (Bio-Rad). The same amounts of total proteins were boiled for 10 min with 1× SDS loading buffer at 100 °C and separated in 8% sodium dodecyl sulfate (SDS)-polyacrylamide gels, followed by immunoblotting detection. Immunoblotting results were visualized using chemiluminescence imaging system (Tanon).

All antibodies used in this study are listed below: anti-MBP (#E8032S, New England Biolabs, 1:5000 dilution), anti-GST (AE006, Abclonal, 1:1000 dilution), anti-Flag (F3165, Sigma-Aldrich, 1:2000 dilution), anti-MYC (C3956, Sigma-Aldrich, 1:2000 dilution), anti-GFP (M20004, Abmart, 1:5000 dilution), anti-COP1 (1:1000 dilution, homemade), anti-Actin (AC009, Abclonal, 1:2000 dilution), anti-Histone H3 (H0164, Sigma-Aldrich, 1:5000 dilution), anti-HSP (AbM51099-31-PU, Beijing Protein Innovation, 1:5000 dilution), anti-pan-Ub (sc-8017, Santa Cruz, 1:1000 dilution), anti-K63-Ub (5621S, Cell Signaling Technology, 1:1000 dilution), and anti-K48-Ub (8081S, Cell Signaling Technology, 1:1000 dilution).

**Expression and purification of recombinant proteins**
The corresponding constructs were transformed into the *Escherichia coli* strain BL21. A small amount of bacterial cells was inoculated into 5 mL LB liquid medium with the appropriate antibiotic and cultured overnight at 37 °C with shaking at 220 rpm. Then 100 μL bacterial cultures were inoculated into 100 mL fresh LB medium and grown at 37 °C with 220 rpm shaking until the optical density (OD$_{600}$) reached 0.5. 1 mM isopropyl-β-D-thiogalactopyranoside (IPTG) was added and the cultures were shaken at 16 °C (120 rpm) for another 16 h to induce the expression of GST-GH3.5, MBP-COP1, MBP-COP1-N, MBP-COP1-C, His-UBC10, His-Ub or His-Ub-K63 proteins. The bacterial cultures were then centrifuged at 3783 × *g* for 7 min at 4 °C and the pellets were collected.

For purification of His-tagged proteins, the pellets were resuspended with 15 mL pre-chilled lysis buffer I (50 mM NaH$_2$PO$_4$.H$_2$O, 300 mM NaCl, 20 mM imidazole, pH 8.0). After sonication and centrifugation at 16,639 × *g* for 30 min at 4 °C, the supernatant was incubated with pre-cleaned 80 μL Ni-NTA agarose beads (QIAGEN) for 2 h at 4 °C. The beads were then collected and washed three times with lysis buffer I. The recombinant proteins were eluted with elution buffer I (50 mM NaH$_2$PO$_4$.H$_2$O, 300 mM NaCl, 250 mM imidazole, 10% [v/v] glycerol, 1× protease inhibitor cocktail, pH 8.0). To quantify the protein concentration, the purified proteins were separated in an SDS-polyacrylamide gel with 1 μg bovine serum albumin (BSA) as reference standard. The gel stained with Coomassie blue was imaged and used to quantify protein concentration using ImageJ software.

The purification of GST- and MBP-tagged proteins was also carried out according to the method described above with some modifications. Bacterial cells expressing GST- or MBP-tagged proteins were resuspended in lysis buffer II (20 mM Tris-HCl [pH 8.0], 300 mM NaCl, 10% [v/v] glycerol) and sonicated. GST-tagged proteins and MBP-tagged proteins were captured with Glutathione Sepharose™ 4B beads (GE Healthcare) and amylose resin (New England Biolabs), respectively. Finally, GST-tagged proteins and MBP-tagged proteins were eluted with elution buffer II (40 mM Tris-HCl [pH 9.0], 300 mM NaCl, 10% [v/v] glycerol, 20 mM GSH, 1× protease inhibitor cocktail) and elution buffer III (40 mM Tris-HCl [pH 9.0], 300 mM NaCl, 10% [v/v] glycerol, 50 mM maltose, 1× protease inhibitor cocktail), respectively.

For the ubiquitination assay, the imidazole-containing elution buffer in which the His-UBC10, His-Ub and His-Ub-K63 proteins were stored was replaced by TBS + 10% glycerol using the Amicon® Pro Purification System (Millipore).

### In vitro pull-down assay

1 μg GST-GH3.5 and 1 μg MBP-COP1 were incubated in 1 mL binding buffer (25 mM Tris-HCl [pH 7.5], 100 mM NaCl, 0.25% NP40) at 4 °C for 1 h. Then, 15 μL amylose resins (New England Biolabs) pre-washed with TBS buffer were added into the binding buffer and incubated for another 1 h. The resins were then washed three times with wash buffer (25 mM Tris-HCl [pH 7.5], 500 mM NaCl, 0.5% NP40), and the proteins were subsequently eluted in 1× SDS loading buffer at 100 °C for 10 min. The eluted proteins were detected by immunoblotting with anti-MBP and anti-GST antibodies.

### Co-IP assay

0.6 g seedlings grown in the dark for 4 d were homogenized in liquid nitrogen and resuspended in Co-IP lysis buffer containing 50 mM Tris-HCl (pH 7.5), 150 mM NaCl, 1 mM EDTA, 10% (v/v) glycerol, 0.05% (v/v) Tween-20, 1 mM PMSF and 1× protease inhibitor cocktail. After centrifugation at $13,000 \times g$ for 10 min at 4 °C, protein concentrations were determined using the Bradford method. 2 mg total proteins were incubated with 15 μL anti-GFP nanobody agarose beads (KTSM1301, AlpalifeBio) in 2 mL Co-IP lysis buffer for 2 h at 4 °C. The anti-GFP beads were then washed three times at 4 °C with 500 μL wash buffer containing 20 mM Tris-HCl (pH 8.0), 150 mM NaCl, 1 mM EDTA, 10% (v/v) glycerol, 0.05% (v/v) Tween-20 and 1× protease inhibitor cocktail. The immunoprecipitated proteins were eluted in 1× SDS loading buffer at 100 °C and analyzed by immunoblotting with anti-GFP, anti-COP1 and anti-actin antibodies.

### LCI assay

The pCAMBIA1300-P19 (suppress RNA silencing)[75], the GH3.5-nLUC, and cLUC-COP1 constructs[71] were individually transformed into the GV3101 strain. GV3101 colonies were inoculated in 2 mL LB medium with gentamycin and kanamycin and cultured overnight at 28 °C with shaking 220 rpm. Then 50 μL cultures were inoculated in 5 mL LB medium containing 10 mM MES (pH 5.6) and 40 μM acetosyringone and grown for 16 h at 28 °C. After centrifugation, the pellets were resuspended in buffer containing 10 mM MES (pH 5.6), 10 mM MgCl₂ and 200 μM acetosyringone. The required bacterial solutions were mixed with a final $OD_{600} = 0.5$ of each bacterium. Subsequently, the bacteria were kept in the dark at room temperature for 3 h and then infiltrated into the fully expanded tobacco leaves using a 1 mL needleless syringe. After 3 d of growth in the dark, the leaves were infiltrated with 1 mM luciferin (E1605, Promega), and the luciferase signal was captured by Night SHADE LB 985 system (Berthold Technologies).

### BiFC assay

For protoplast isolation, the healthy and well-developed leaves of 4-week-old *Arabidopsis* plants were selected. Protoplasts were isolated from 0.5 to 1 mm wide leaf strips in the freshly prepared enzyme solution containing 20 mM MES (pH 5.7), 0.4 M mannitol, 20 mM KCl, 1.5% (w/v) cellulase R10, 0.4% (w/v) macerozyme R10, 10 mM CaCl₂, 0.1% (w/v) BSA[76]. After enzymolysis in the dark for 3 h at room temperature with slow shaking (60 rpm), the solution was filtered with a nylon mesh to remove undigested leaf tissue. Then the protoplasts were collected by centrifugation at $100 \times g$ for 2 min and washed twice with 5 mL W5 solution containing 2 mM MES (pH 5.7), 154 mM NaCl, 125 mM CaCl₂, and 5 mM KCl. The protoplasts were then resuspended in MMG solution containing 4 mM MES (pH 5.7), 0.4 M mannitol, 15 mM MgCl₂. 200 μL protoplasts were gently mixed with 10 μg of each plasmid. Then 250 μL of PEG-calcium solution containing 0.2 M mannitol, 100 mM CaCl₂, 40% (w/v) PEG4000 was added and mixed gently. After 20 min of transfection at room temperature, 1 mL W5 solution was added to stop the transfection. The protoplasts were collected and resuspended in 1 mL WI solution containing 4 mM MES (pH 5.7), 0.5 M mannitol, 20 mM KCl. After 14 h of culture in the dark at 22 °C, the protoplasts were then mounted on microscope slides and the fluorescence signals of YFP and mCherry were detected using a Zeiss LSM880 confocal microscope (Carl Zeiss). YFP fluorescence was excited with a 514-nm laser, and the emission spectra were recorded from 519 to 603 nm. The mCherry fluorescence was excited with a 561-nm laser, and the emission spectra were recorded from 590 to 628 nm.

*Arabidopsis GH3.5-GFP* seedlings grown in continuous darkness for 4 d were soaked in 50 μM 4′,6-diamidino-2-phenylindole dihydrochloride (DAPI, D9542, Sigma-Aldrich) solution. After staining for 5 min, the finished samples were mounted on glass slides. Images were captured using a confocal microscope. The GFP signal was excited at 488 nm and the emission spectra were recorded in a range from 500 to 550 nm. The DAPI signal was excited at 405 nm and the emission spectra were recorded in a range from 425 to 475 nm.

### Quantitative real-time PCR

Total RNA was isolated using the Eastep® Super Total RNA Extraction Kit (LS1040, Promega). For reverse transcription, 1 μg total RNA was used as a template to synthesize first-strand cDNAs using the Hifair® III 1st Strand cDNA Synthesis SuperMix for qPCR (11141ES60, Yeasen). The RT-qPCR assay was performed using the Hieff UNICON® Universal Blue qPCR SYBR Green Master Mix (11184ES08, Yeasen) with the ABI QuantStudio™ 6 Flex Real-Time PCR System (Thermo Fisher Scientific). The relative expression levels of different genes were analyzed using the $2^{-\Delta\Delta Ct}$ method and normalized to those of *PP2A*. All primers for RT-qPCR assay are listed in Supplementary Table 1.

### In vitro ubiquitination assay

60 μL ubiquitination reaction mixtures contained 50 nM His-UBE1 (#E-304-050, Boston Biochem), 100 ng His-UBC10, 1 μg COP1-SPA4 complex[77], 4 μg His-Ub and 330 ng GST-GH3.5 in a reaction buffer containing 50 mM Tris-HCl (pH 7.5), 5 mM MgCl₂, 2 mM ATP, and 10 μM ZnCl₂. After incubation for 2 h at 30 °C, the reaction was terminated by addition of 5× SDS loading buffer. The polyubiquitination of GH3.5 was detected with anti-GST and anti-pan-Ub antibodies.

### In vivo ubiquitination assays

For the transient expression experiment, *Arabidopsis* protoplasts containing different vector combinations were incubated in the dark for 12 h. Total proteins were extracted with Co-IP lysis buffer with the addition of various deubiquitinating enzyme (DUB) inhibitors, including 5 mM N-ethylmaleimide (NEM, Sigma-Aldrich), 5 mM 1,10-phenanthroline monohydrate (Sigma-Aldrich), and 50 μM 2,6-diaminopyridine-3,5-bis(thiocyanate) (PR-619, Sigma-Aldrich). GH3.5-GFP proteins were immunoprecipitated with anti-GFP agarose beads (KTSM1301, AlpalifeBio) at 4 °C. Then the anti-GFP beads were washed three times with lysis buffer containing DUB inhibitors. Ubiquitination of GH3.5-GFP was detected with the corresponding antibodies.

For the ubiquitination assay *in planta*, 4-day-old dark-grown Col, *GH3.5-GFP*, and GH3.5-GFP/*cop1-4* seedlings were used to immuno-precipitate GH3.5-GFP proteins with GFP-trap magnetic agarose beads. Polyubiquitination of GH3.5-GFP was detected with anti-pan-Ub, anti-K63 Ub, anti-K48 Ub and anti-GFP antibodies.

### GH3.5 enzymatic activity experiment and kinetic analysis

To test the enzymatic activity of GH3.5, GST-GH3.5 proteins linked to K63 ubiquitin chains (GST-GH3.5-K63) were generated by a COP1-mediated in vitro ubiquitination assay. In parallel, an in vitro ubiqui-tination reaction was performed without GST-GH3.5. After completion of the reaction, GST-GH3.5 proteins were added into the mixture, which was considered as GST-GH3.5 without ubiquitin chains (GST-GH3.5). To enrich GST-GH3.5 and GST-GH3.5-K63 proteins, the ubi-quitination mixtures were incubated with GST beads in 1 mL TBS buffer at 4 °C for 1 h. Subsequently, the beads were washed three times with TBS buffer at 4 °C and used to catalyze IAA to IAA-Asp. The beads were mixed with 200 μL reaction buffer containing 50 mM Tris-HCl (pH 8.0), 2 mM $MgCl_2$, 0.15 mM ATP, 1 mM DTT, 10 mM Asp, and 2 mM IAA to start the reaction. The reactions were carried out at 25 °C with gentle rotation on a vertical mixer. After incubation for 30 min, each reaction was quenched by adding 7 μL concentrated hydrochloric acid. An equal volume of ethyl acetate was mixed with the reaction solution on a vortex. The mixture was then centrifuged at $13,000 \times g$ for 10 min at 4 °C, and the supernatant was transferred into a new tube. The liquid-liquid extraction was repeated three times. The supernatants were combined and evaporated to dryness at room temperature. Then 100 μL methanol was added to redissolve the solutes containing IAA-Asp. The production of IAA-Asp was quantified by ultra-performance liquid chromatography tandem mass-spectrometry (UPLC–MS/MS) system (Thermo Vanquish UHPLC system; Q Exactive mass spectro-meter; Thermo Scientific, USA)[44]. Gradient elution was performed with solvent A (water with 0.1% formic acid) and solvent B (methanol). The mobile phase was delivered at a flow rate of 0.3 mL/min under the following gradient conditions: 0-0.3 min: 0–5% B, 0.3-1 min: 5–41% B, 1–4 min: 41–45% B, 4–4.2 min: 45–80% B, 4.2–4.5 min: 80-100% B, holding at 100% B for 2 min, then back to 5% B in 0.2 min and holding at 5% B for 1.3 min. The effluent was introduced into the mass spectro-meter with the following optimal settings: positive ionization mode; spray voltage, 3800 V; capillary temp, 320 °C; sheath gas flow, 40 Arb; auxiliary gas flow, 11 Arb; aux gas heater temp, 370 °C; scan range, 100–350 m/z. Three independent replicates were used for each treat-ment. Raw data were analyzed using Xcalibur 2.1 (Thermo Scientific, USA) with default settings. The peak of IAA-Asp was detected in the mass range (m/z 291.09495–291.09787).

The kinetic parameters were determined through initial velocity experiments[78]. Before analyzing the steady-state kinetics, the condi-tions under which GST-GH3.5-catalyzed production of IAA-Asp was linear over time were determined. Subsequently, the enzymatic reac-tion, which was catalyzed by approximately 125 ng GST-GH3.5, was carried out at various IAA concentrations (0.004, 0.008, 0.02, 0.05, 0.2, and 0.5 mM) for 5 min to calculate the initial velocities. The pro-duction of IAA-Asp was quantified using UPLC–MS/MS[44]. Initial velo-cities as a function of IAA concentrations were fitted to the Michaelis–Menten equation using GraphPad Prism software (version 8.0). The parameters for the UPLC-MS/MS analyzes are listed in Sup-plementary Table 2.

### Measurement of IAA and IAA metabolites

The contents of endogenous IAA, IAA-Asp, IAA-Glu, and oxIAA-Asp were determined by the Wuhan Greensword Creation Technology Company (http://www.greenswordcreation.com) with UHPLC-MS. 0.1 g of 4-day-old *Arabidopsis* seedlings grown in continuous darkness or continuous white light (35 μmol $m^{-2}$ $s^{-1}$) were harvested, frozen in liquid nitrogen and ground into fine powder. Three to five biological replicates were prepared. Each sample was extracted with 1.0 mL 80% methanol (v/v) at 4 °C for 12 h and then centrifuged at $10,000 \times g$ for 10 min at 4 °C. The supernatant was collected and evaporated under mild nitrogen stream at 35 °C followed by re-dissolving in 100 μL $H_2O$/ACN (90/10, v/v) for UHPLC-MS/MS analysis (Thermo Scientific Ulti-mate 3000 UHPLC coupled with TSQ Quantiva). Formic acid in water (0.1%, v/v, solvent A) and ACN (solvent B) were employed as mobile phases at a flow rate of 0.4 mL/min. A gradient of 0−8 min 5%−40% B, 8−9 min 40−90% B, 9−12 min 90% B, 12−12.5 min 90−5% B, and 12.5−15 min 5% B was employed. For MS analysis, ion source conditions were as follows: Sheath gas flow, 35 Arb; auxiliary gas flow, 10 Arb; sweep gas flow, 8 Arb; collision gas, 1.5 mTorr; ion transfer tube tem-perature, 350 °C; vaporizer temperature, 300 °C; spray voltage, 3.5 kV. Data acquisition and analysis were performed using a Thermo Scien-tific Xcalibur 2.1 data system. The parameters for the UPLC-MS/MS analyzes are listed in Supplementary Table 2.

### Primer sequences

All primer sequences used in this study are listed in Supplementary Table 1.

### Statistics and reproducibility

The statistical analyzes in this study were performed with GraphPad Prism (version 8.0) and provided in the Source Data file. *P* values and different lowercase letters above the histogram represent statistically significant differences between groups. For hypocotyl length mea-surements, *n* indicates the number of seedlings used in the experi-ment. BiFC, LCI and western blot experiments were performed at least twice with similar results.

### Reporting summary

Further information on research design is available in the Nature Portfolio Reporting Summary linked to this article.

## Data availability

All data supporting the findings of this study are available within the paper and its Supplementary Information. Source data are provided with this paper.

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

## Acknowledgements

We are grateful to Prof. Wei Xiao (Capital Normal University) for providing the *ubc35-1 ubc36-1* mutant, Prof. Yunde Zhao (University of California, San Diego) for providing the *gh3(5/6/9/17)* and *gh3-septuple* mutants, Prof. Hongwei Guo (Southern University of Science and Technology) and Prof. Kai Jiang (Yunnan University) for providing the chemical inhibitor nalacin, Prof. Ping Yin (Huazhong Agricultural University) for providing the recombinant COP1-SPA4 complex, and Prof. Jigang Li (China Agricultural University) for providing the pCAMBIA1307-*35S:HF-COP1* construct. We thank Ms. Suying Gao (Southern University of Science and Technology) for her help with UPLC-MS experiments. We thank LetPub (www.letpub.com.cn) for its linguistic assistance during the preparation of this manuscript. This work was supported by grants from the National Key R&D Program of China (Grant No. 2024YFA1306701 to X.W.D.), the National Natural Science Foundation of China (Grant No. 32200198 to J.L.), the Priority Academic Program Development of Jiangsu Higher Education Institutions (PAPD), the Shandong Provincial Natural Science Foundation (Grant No. SYS202206 and ZR2021ZD30 to X.W.D.), Southern University of Science and Technology (Y01226026 to X.W.D.), and the Shenzhen Science and Technology Program (Grant No. ZDSYS20230626091659010).

## Author contributions

X.W.D., and J.L. conceived the project and designed experiments; Y.L., Y.X., D.X., and J.L. performed experiments; Y.L., Y.X., X.W.D., and J.L. analyzed data; Y.L., X.W.D., and J.L. wrote the paper.

## Competing interests

The authors declare no competing interests.
