## [Transparent Peer Review file · Nature Communications]

Inactivation of GH3.5 by COP1-mediated K63-linked ubiquitination promotes seedling hypocotyl elongation

Corresponding Author: Professor Jian Li

Version 0:

Reviewer comments:

Reviewer #1

(Remarks to the Author)

These authors show that inactivation of GH3.5 by COP1-mediated K63-linked ubiquitination promotes seedling hypocotyl elongation. The authors used very solid in vitro and in vivo data to show that COP1 physically interacts with GH3.5 to inhibit its enzyme activity through K63 ubiquitination. Also, the authors found that light regulates IAA metabolism by inhibiting COP1 function. This story expands our understanding of COP1 function in plants especially its non-proteasomal role in regulating plant development. A few suggestions to improve the story:

1. Line 115 in introduction states "Overexpression of different GH3s show short hypocotyls and a dwarf phenotype in the light". However, Fig. 1A shows higher order gh3 mutants also exhibit short hypocotyls in the dark. It appears GH3 exhibits opposite function in hypocotyl elongation under light and dark conditions. Can the authors explain these data?
2. In Figures 6, authors show that light regulates IAA metabolism by suppressing COP1 function. They should include phenotypes of the seedlings under the same conditions or at least dark and light to test if these changes are meaningful at the physiological level.
3. Minor change, Figure 4a. The "+" sizes are different in different lanes.

Reviewer #2

(Remarks to the Author)

This manuscript describes a comprehensive mechanism of COP1-mediated K63-linked ubiquitination regulates hypocotyl elongation. This study provides very interesting findings and will be of great interest to plant biology field. The manuscript is carefully designed and most of the necessary controls are present. I only have few comments:

1. In Fig. 4c, the data requires negative controls, GH3.5-GFP/MYC-Ub-WT and GH3.5-GFP/MYC-Ub-K63R.
2. Nalacin inhibits the primary root growth of seedlings by inhibiting GH3 activity. Why nalacin could not inhibit the primary root growth of cop1-4 to the wild-type level? (Fig. 5b). In other words, why GH3 in cop1-4 is resistant to nalacin?
3. In cop1-4, the GH3 activity is higher than Col-0. Thus, it would be expected that IAA levels in cop1-4 should be lower than that of wild-type under darkness. But the results showed that cop1-4 accumulated slightly higher level of IAA under darkness (Fig. 5g and 6b). Please explain why? Moreover, if cop1-4 really accumulates higher level of IAA under darkness, the hypocotyl length of cop1-4 should be longer than Col-0. But the cop1-4 exhibits short hypocotyl phenotype under darkness. Please explain why?

Reviewer #3

(Remarks to the Author)

CONSTITUTIVELY PHOTOMORPHOGENIC 1 (COP1) is an E3 ubiquitin ligase that plays a central role in light signaling and seedling development under both light and dark conditions. The current model posits that COP1 mediates the K48-

linked polyubiquitination for the degradation of target proteins, including many transcription factors such as HY5 and HFR1. Here, the authors provide strong evidence supporting that COP1 can also mediate K63-linked polyubiquitination, which modifies the target protein's activity instead of proteolysis. The authors showed that one target protein of the COP1 K63 ubiquitin ligase activity is GRETCHEN HAGEN 3.5 (GH3.5), which is an auxin modification enzyme that conjugates amino acids to indole-3-acetic acid (IAA). COP1 interacts directly with GH3.5 to attenuate the activity of GH3.5 via K63 polyubiquitination, thereby enhancing the level of free auxin to promote hypocotyl elongation in the dark.

Overall, the study reveals novel insights into the function of COP1 as an E3 ubiquitin ligase for K63-linked polyubiquitination and draws a new connection between COP1 and the regulation of amino-acid-conjugated auxin. The manuscript is well written. The conclusions are supported by solid genetic and biochemical evidence. I have only one suggestion that hopefully will help improve the manuscript for publication.

– Line 177: “The biomolecular fluorescence complementation assay in Arabidopsis protoplasts showed a clear interaction between COP1 and GH3.5 in the nucleus (Fig. 2d). Also, the final model (Fig. 7) indicates that COP1 modulates the activity of GH3.5 and amino-acid-conjugation of IAA in the nucleus. GH3.5 and amino-acid-conjugation of auxin have been studied previously. In the context of previous studies, please discuss whether the subcellular localization of the COP1-GH3.5 regulation is aligned with the current knowledge of amino-acid-conjugation of IAA.

Reviewer #4

(Remarks to the Author)

In this submission by Liu and colleagues, the role of non-proteolytic ubiquitination of GH3.5 is found to be mediated by the E3 ligase COP1. Furthermore, this post-translational modification of GH3.5 inhibits its activity, promoting hypocotyl elongation in the dark, which is a noteworthy and original finding. Overall this is an interesting study, but I have a few concerns about the kinetics data and the oversight of the products that are formed by GH3.5. My comments are as follows:

Major comments:

1. The Michaelis-Menten kinetics plots in Figure 5f are an inaccurate representation of the data. Because this is an endpoint assay (reaction stopped after 5 minutes) and not a steady-state kinetics experiment, maximum velocity (V_{max}) cannot be reported. However, maximal activity at 0.5 mM substrate can be accurately reported for comparison purposes. The values on the y-axis report units of nmol/min/mg, but since every reaction was conducted over 5 minutes, the rate value is missing, so this should be nmol/mg to more accurately represent the data. Others have used well established continuous kinetics assays that can be done in a 96-well plate using a spectrophotometer, and this would give rate data that could be used to report V_{max} and turnover.
2. In the in vitro assays, ATP concentrations were only 0.1 mM, which may be limiting at substrate concentrations above 0.1 mM.
3. The kinetics data should be conducted in triplicate, which is unclear from the singlet points and absence of error bars on the plot in Figure 5f.
4. GH3.5 is promiscuous and can conjugate amino acids onto multiple auxins (IAA and phenylacetic acid) and benzoates (salicylic acid and benzoic acid) in vitro and in vivo. In this publication (www.pnas.org/cgi/doi/10.1073/pnas.1612635113), GH3.5 has a higher catalytic efficiency with BA relative to IAA. Perhaps it would be worth measuring BA-Asp, PAA-Asp, and SA-Asp to determine whether other products of GH3.5 are affected by light and dark.
5. In Figure 5a, the 0 mM data should also be included for comparison.
6. More information should be provided in the methods regarding protein expression, purification, the buffers that were used, and how protein concentration was determined.

Minor point:

1. On lines 255 and 258: “catalytic rate” is used to describe the kinetics, which will need to be modified unless steady-state, continuous assays are used instead of endpoint assays.

Version 1:

Reviewer comments:

Reviewer #1

(Remarks to the Author)

My concerns have been satisfactorily addressed.

Reviewer #2

(Remarks to the Author)

The authors have thoroughly addressed all my comments. I have no further suggestions for this MS.

Reviewer #3

(Remarks to the Author)

The authors have nicely addressed my comments.

Reviewer #4

(Remarks to the Author)

The additions to the methods and clarifications that were made to the figures and legends have greatly improved the clarity of the kinetics experiments, and I have no additional concerns.

Reviewer's Comments:

Reviewer #1 (Remarks to the Author)

These authors show that inactivation of GH3.5 by COP1-mediated K63-linked ubiquitination promotes seedling hypocotyl elongation. The authors used very solid in vitro and in vivo data to show that COP1 physically interacts with GH3.5 to inhibit its enzyme activity through K63 ubiquitination. Also, the authors found that light regulates IAA metabolism by inhibiting COP1 function. This story expands our understanding of COP1 function in plants especially its non-proteasomal role in regulating plant development. A few suggestions to improve the story:

1. Line 115 in introduction states “Overexpression of different GH3s show short hypocotyls and a dwarf phenotype in the light”. However, Fig. 1A shows higher order *gh3* mutants also exhibit short hypocotyls in the dark. It appears GH3 exhibits opposite function in hypocotyl elongation under light and dark conditions. Can the authors explain these data?

Response:

Thank you for the comment. Moderate auxin levels promote hypocotyl elongation. However, excessive activation of the auxin pathway, including both excessive auxin biosynthesis and signaling, has been reported to inhibit hypocotyl elongation in the dark^{1,2,3}. Knockout of four group II *GH3s* (*GH3.5*, *GH3.6*, *GH3.9*, and *GH3.17*) in Col results in excessive activation of endogenous auxin by inhibiting the conversion of IAA to IAA-amino acid conjugates⁴, thereby impairing hypocotyl elongation in the dark. To further investigate, we applied Kyn + PPBo^{5,6}, two inhibitors of auxin biosynthesis, to Col and *gh3(5/6/9/17)* seedlings in the dark to downregulate endogenous auxin synthesis. As shown in Fig. R1, the hypocotyls of *gh3(5/6/9/17)* were dramatically longer than those of Col. Moreover, the concentration of inhibitors used in the experiment even slightly promoted hypocotyl elongation of etiolated *gh3(5/6/9/17)*,

which is consistent with the previous reports that moderate inhibition of excessive IAA accumulation promotes cell elongation in the dark³. In summary, GH3s inactivate auxin to inhibit hypocotyl elongation. Excessive activation of endogenous auxin in the etiolated higher-order *gh3* mutants contributes to their short hypocotyls of these mutants in the dark.

Fig. R1 *gh3(5/6/9/17)* is insensitive to inhibitors of auxin biosynthesis. Hypocotyl lengths of Col and *gh3(5/6/9/17)* seedlings treated with 15 μ M Kyn + 1 μ M PPBo, two inhibitors of auxin biosynthesis. DMSO was the solvent for the inhibitors. Data are presented as the mean \pm SD, $n = 30$. Different lowercase letters above the histogram represent statistically significant differences ($P < 0.0001$) determined by ordinary two-way ANOVA with Tukey's post hoc test.

The references mentioned in the response are listed below:

1. Zhao Y, *et al.* A role for flavin monooxygenase-like enzymes in auxin biosynthesis. *Science* **291**, 306-309 (2001).
2. Gray WM, *et al.* Identification of an SCF ubiquitin–ligase complex required for auxin response in *Arabidopsis thaliana*. *Genes Dev* **13**, 1678-1691 (1999).
3. Du M, *et al.* Biphasic control of cell expansion by auxin coordinates etiolated seedling development. *Sci Adv* **8**, eabj1570 (2022).
4. Guo R, *et al.* Local conjugation of auxin by the GH3 amido synthetases is required

for normal development of roots and flowers in Arabidopsis. *Biochem Biophys Res Commun* **589**, 16-22 (2022).

5. He W, *et al.* A small-molecule screen identifies L-kynurenine as a competitive inhibitor of TAA1/TAR activity in ethylene-directed auxin biosynthesis and root growth in Arabidopsis. *Plant Cell* **23**, 3944-3960 (2011).

6. Kakei Y, *et al.* Small-molecule auxin inhibitors that target YUCCA are powerful tools for studying auxin function. *Plant J* **84**, 827-837 (2015).

2. In Figures 6, authors show that light regulates IAA metabolism by suppressing COP1 function. They should include phenotypes of the seedlings under the same conditions or at least dark and light to test if these changes are meaningful at the physiological level.

Response:

Thank you for your suggestion. We have analyzed the phenotypes of seedlings grown under continuous darkness or continuous white light. As shown in Supplementary Figure 5 (see below), light-inhibited hypocotyl elongation was significantly attenuated in the *gh3(5/6/9/17)* mutant, which was consistent with the reduced production of IAA-Asp + oxIAA-Asp in *gh3(5/6/9/17)* in the light (Fig. 6e). Moreover, knockout of the four *GH3s* in *cop1-4* partially restored hypocotyl elongation in *cop1-4* under light conditions (Supplementary Figure 5), also indicating an inverse correlation between hypocotyl elongation and IAA-Asp + oxIAA-Asp accumulation in *cop1-4* (Fig. 6e). These results suggest that light modulates hypocotyl elongation partially by regulating IAA metabolism.

Supplementary Fig. 5 Light inhibits hypocotyl elongation by attenuating *COPI*-mediated inhibition of *GH3s* function. a Phenotypes of seedlings of Col, *cop1-4*, and *gh3* mutants in Col and *cop1-4* backgrounds. Seedlings were grown in continuous darkness (cD) or continuous white light (cWL, $35 \mu\text{mol m}^{-2} \text{s}^{-1}$) for 4 days. Scale bar, 1 mm. **b** Hypocotyl lengths of seedlings as indicated in (a). Data are presented as the mean \pm SD, $n = 30$. Different lowercase letters above the histogram represent statistically significant differences ($P < 0.0001$) determined by ordinary two-way ANOVA with Tukey's post hoc test.

3. Minor change, Figure 4a. The "+" sizes are different in different lanes.

Response:

Thank you for the reminder. We have standardized the "+" sizes across all lanes in Figure 4a.

Reviewer #2 (Remarks to the Author)

This manuscript describes a comprehensive mechanism of COP1-mediated K63-linked ubiquitination regulates hypocotyl elongation. This study provides very interesting findings and will be of great interest to plant biology field. The manuscript is carefully designed and most of the necessary controls are present. I only have few comments:

1. In Fig. 4c, the data requires negative controls, GH3.5-GFP/MYC-Ub-WT and GH3.5-GFP/MYC-Ub-K63R.

Response:

Thank you for the suggestion. We have repeated the experiment as recommended. Consistent with the earlier version, the new Fig. 4c (see below) showed that K63R mutant ubiquitin expression resulted in impaired ubiquitination of GH3.5 by COP1 in protoplasts, indicating that COP1 mainly attaches K63 ubiquitin chains to GH3.5 in vivo.

Fig. 4c COP1-mediated ubiquitination of GH3.5 with the co-expression of the

K63R mutant ubiquitin in protoplasts. MYC-Ub-WT, MYC-tagged wild-type ubiquitin. MYC-Ub-K63R, MYC-tagged ubiquitin with Lys⁶³ substituted by arginine (R). Ubiquitination of immunoprecipitated GH3.5-GFP proteins was detected with an anti-MYC antibody.

2. Nalacin inhibits the primary root growth of seedlings by inhibiting GH3 activity. Why nalacin could not inhibit the primary root growth of *cop1-4* to the wild-type level? (Fig. 5b). In other words, why GH3 in *cop1-4* is resistant to nalacin?

Response:

Thank you for the comment. Nalacin likely inhibits GH3 activity by occupying the IAA binding sites on GH3s¹. Our in vitro kinetic analysis indicates that COP1-mediated K63 ubiquitination does not affect the affinity of GH3.5 for IAA (Fig. 5f), which may limit the inhibitory effect of nalacin on the increased GH3 activity in *cop1-4*. Nevertheless, due to the inhibitory mechanism of nalacin, it can be assumed that GH3s activity in *cop1-4* is fully suppressed when a sufficient amount of nalacin occupies the IAA binding sites on all GH3 molecules. Indeed, when the nalacin concentration was increased to 24 μ M, root growth inhibition in *cop1-4* was comparable to that in Col (Fig. R2). Investigating the structural mechanism underlying the inhibition of GH3.5 activity by COP1-mediated ubiquitination will be a valuable direction for future studies.

Fig. R2 Nalacin inhibition of primary root growth in Col and *cop1-4*. Col and *cop1-4* seedlings were grown in the dark on MS medium supplemented with 12 or 24 μ M

nalacin. Root growth inhibition is expressed as the percentage decrease in length relative to seedlings grown without nalacin. Each data point represents the average values of 25 seedlings. Different lowercase letters above the histogram represent statistically significant differences ($P < 0.001$) determined by ordinary two-way ANOVA with Tukey's post hoc test.

The reference mentioned in the response is listed below:

1. Xie Y, *et al.* Chemical genetic screening identifies nalacin as an inhibitor of GH3 amido synthetase for auxin conjugation. *Proc Natl Acad Sci U S A* **119**, e2209256119 (2022).

3. In *cop1-4*, the GH3 activity is higher than Col-0. Thus, it would be expected that IAA levels in *cop1-4* should lower than that of wild-type under darkness. But the results showed that *cop1-4* accumulated slightly higher level of IAA under darkness (Fig. 5g and 6b). Please explain why? Moreover, if *cop1-4* really accumulates higher level of IAA under darkness, the hypocotyl length of *cop1-4* should longer than Col-0. But the *cop1-4* exhibits short hypocotyl phenotype under darkness. Please explain why?

Response:

Thank you for the comment. Although the free IAA content was slightly higher in *cop1-4* under dark conditions than in Col, the difference was not statistically significant (Figs. 5g and 6b). Our study shows nuclear localization of COP1-GH3.5 regulation (Fig. 2d, Fig. 7, Supplementary Fig. 6 in the revised manuscript). Since we measured endogenous IAA content in whole seedlings, future work should investigate whether nuclear IAA content decreases locally in etiolated *cop1-4* seedlings.

In this study, although the total free IAA content in *cop1-4* was comparable to that in Col, the content of IAA-Asp and its oxidized form (IAA-Asp + oxIAA-Asp) was substantially higher in etiolated *cop1-4* than in Col (Fig. 5g and 6e). Knockout of four group II *GH3s* (including *GH3.5*, *GH3.6*, *GH3.9*, and *GH3.17*) in etiolated *cop1-4* inhibited the excessive IAA-Asp + oxIAA-Asp accumulation in *cop1-4* (Fig. 6e). These results suggest increased GH3s activity in *cop1-4* in the dark. Moreover, the impaired

hypocotyl elongation of etiolated *cop1-4* was partially restored by the loss of *GH3s* function (Fig. 1), indicating an inverse correlation between hypocotyl elongation and IAA-Asp + oxIAA-Asp accumulation in *cop1-4*. Taken together, these results suggest that abnormal auxin metabolism in *cop1-4* contributes to the short hypocotyl of *cop1-4* in the dark.

Reviewer #3 (Remarks to the Author)

CONSTITUTIVELY PHOTOMORPHOGENIC 1 (COP1) is an E3 ubiquitin ligase that plays a central role in light signaling and seedling development under both light and dark conditions. The current model posits that COP1 mediates the K48-linked polyubiquitination for the degradation of target proteins, including many transcription factors such as HY5 and HFR1. Here, the authors provide strong evidence supporting that COP1 can also mediate K63-linked polyubiquitination, which modifies the target protein's activity instead of proteolysis. The authors showed that one target protein of the COP1 K63 ubiquitin ligase activity is GRETCHEN HAGEN 3.5 (GH3.5), which is an auxin modification enzyme that conjugates amino acids to indole-3-acetic acid (IAA). COP1 interacts directly with GH3.5 to attenuate the activity of GH3.5 via K63 polyubiquitination, thereby enhancing the level of free auxin to promote hypocotyl elongation in the dark.

Overall, the study reveals novel insights into the function of COP1 as an E3 ubiquitin ligase for K63-linked polyubiquitination and draws a new connection between COP1 and the regulation of amino-acid-conjugated auxin. The manuscript is well written. The conclusions are supported by solid genetic and biochemical evidence. I have only one suggestion that hopefully will help improve the manuscript for publication.

– Line 177: “The biomolecular fluorescence complementation assay in *Arabidopsis* protoplasts showed a clear interaction between COP1 and GH3.5 in the nucleus (Fig. 2d). Also, the final model (Fig. 7) indicates that COP1 modulates the activity of GH3.5

and amino-acid-conjugation of IAA in the nucleus. GH3.5 and amino-acid-conjugation of auxin have been studied previously. In the context of previous studies, please discuss whether the subcellular localization of the COP1-GH3.5 regulation is aligned with the current knowledge of amino-acid-conjugation of IAA.

Response:

Thank you for your suggestion. We have included a discussion on the subcellular localization of COP1-GH3.5 regulation in the Discussion section of the revised manuscript. The revised text is as follows:

Our results showed an interaction between COP1 and GH3.5 in the nucleus (Fig. 2d), indicating that COP1 modulates GH3.5 activity through K63 ubiquitination in the nucleus (Fig. 7). However, the subcellular localization of group II GH3s in *Arabidopsis* remains unclear³⁸. The GH3s of several other species that produce IAA-amino acid conjugates have been reported to be localized in both the cytoplasm and the nucleus^{48, 49, 50}. Moreover, the GFP fluorescence signals from the GH3.5-GFP/Col line indicate that the GH3.5 proteins are localized in both the cytoplasm and nucleus (Fig. 3, Supplementary Fig. 6), supporting the nuclear interaction between COP1 and GH3.5 (Fig. 2d, Fig. 7). Classical auxin perception and signaling transduction by TRANSPORT INHIBITOR RESPONSE1/AUXIN SIGNALING F-BOX PROTEIN (TIR1/AFB) receptors occur in the nucleus, and are critical for regulating gene expression and plant growth and development^{51, 52}. It is likely that the nuclear localization of COP1-GH3.5 regulation provides a rapid mechanism to modulate the nuclear auxin pathway to regulate gene expression.

Supplementary Fig. 6 Subcellular localization of GH3.5 in vivo. *GH3.5-GFP*

seedlings were grown in continuous darkness for 4 d. The fluorescence signals from the root cells were monitored. DAPI was used to label the nucleus. Scale bar, 10 μm .

The references mentioned in the response are listed below:

38. Wojtaczka P, Ciarkowska A, Starzynska E, Ostrowski M. The GH3 amidosynthetases family and their role in metabolic crosstalk modulation of plant signaling compounds. *Phytochemistry* **194**, (2022).
48. ZHANG S, *et al.* The auxin response factor, OsARF19, controls rice leaf angles through positively regulating 3-5 and 1. *Plant, Cell Environ* **38**, 638-654 (2015).
49. Jiang L, *et al.* MdGH3.6 is targeted by MdMYB94 and plays a negative role in apple water-deficit stress tolerance. *Plant J* **109**, 1271-1289 (2022).
50. Ai G, *et al.* SlGH3.15, a member of the GH3 gene family, regulates lateral root development and gravitropism response by modulating auxin homeostasis in tomato. *Plant Sci* **330**, 111638 (2023).
51. Middleton AM, *et al.* Data-driven modeling of intracellular auxin fluxes indicates a dominant role of the er in controlling nuclear auxin uptake. *Cell Rep* **22**, 3044-3057 (2018).
52. Tang W, Yu Y, Xu T. The interplay between extracellular and intracellular auxin signaling in plants. *J Genet Genomics*, (2024).

Reviewer #4 (Remarks to the Author)

In this submission by Liu and colleagues, the role of non-proteolytic ubiquitination of GH3.5 is found to be mediated by the E3 ligase COP1. Furthermore, this post-translational modification of GH3.5 inhibits its activity, promoting hypocotyl elongation in the dark, which is a noteworthy and original finding. Overall this is an interesting study, but I have a few concerns about the kinetics data and the oversight of the products that are formed by GH3.5. My comments are as follows:

Major comments:

1. The Michaelis-Menten kinetics plots in Figure 5f are an inaccurate representation of the data. Because this is an endpoint assay (reaction stopped after 5 minutes) and not a steady-state kinetics experiment, maximum velocity (V_{max}) cannot be reported. However, maximal activity at 0.5 mM substrate can be accurately reported for comparison purposes. The values on the y-axis report units of nmol/min/mg, but since every reaction was conducted over 5 minutes, the rate value is missing, so this should be nmol/mg to more accurately represent the data. Others have used well established continuous kinetics assays that can be done in a 96-well plate using a spectrophotometer, and this would give rate data that could be used to report V_{max} and turnover.

Response:

Thank you for your comment. In our *in vitro* GH3.5-catalyzed reaction, the ubiquitinated GST-GH3.5 were enriched by GST beads. The reactions were carried out with gentle rotation in a vertical mixer to prevent the beads from sinking to the bottom of the tubes. This setup precluded performing the kinetic assay in a 96-well plate with a spectrophotometer.

Prior to conducting the experiment shown in Fig. 5f, we first tested under which conditions the enzymatic reactions remained in steady state, following a previously published report¹. Reactions catalyzed by GST-GH3.5 with various IAA concentrations (0.05 mM, 0.2 mM, or 0.5 mM) were quenched at different time points (Fig. R3). The production of IAA-Asp was quantified by UPLC-MS. As shown in Fig. R3, IAA-Asp production increased linearly over time within 10 min, indicating a steady-state period for all reactions. Therefore, the initial velocity was calculated from the slope of the linear portion of the product-time curve. Subsequently, the enzymatic reaction catalyzed by GST-GH3.5 was carried out at various IAA concentrations for 5 min to calculate the initial velocities. The initial velocities as a function of IAA concentrations were fitted to the Michaelis-Menten equation (Fig. 5f), from which we were able to derive approximately steady-state kinetic parameters. The test for analyzing the linear relationship between IAA-Asp production and reaction time is described in the

Methods section of the revised manuscript.

Fig. R3 Time-dependent production of IAA-Asp by GST-GH3.5 in vitro. Reactions catalyzed by GST-GH3.5 with various IAA concentrations were quenched at different time points. The linear relationship between IAA-Asp production and reaction time was analyzed through simple linear regression.

The reference mentioned in the response is listed below:

1. Chen Q, Zhang B, Hicks LM, Wang S, Jez JM. A liquid chromatography–tandem mass spectrometry-based assay for indole-3-acetic acid–amido synthetase. *Anal Biochem* **390**, 149-154 (2009).
2. In the in vitro assays, ATP concentrations were only 0.1 mM, which may be limiting at substrate concentrations above 0.1 mM.

Response:

Thank you for the comment. In Fig. 5f, 125 ng GST-GH3.5 proteins were used to catalyze the formation of IAA-Asp. GST-GH3.5 produced the maximum amount (about 0.068 nmol) with 0.15 mM ATP and 0.2 mM IAA as substrates. The total reaction volume was 50 μ l, so the initial ATP amount was calculated to be 7.5 nmol. Additionally, the 0.068 nmol production of IAA-Asp indicated that out of 7.5 nmol ATP pool, only 0.068 nmol ATP was consumed at the end of the reaction, indicating a sufficient amount of ATP in the reaction. Therefore, it is unlikely that the ATP concentration used limits the production of IAA-Asp in the reaction. We have added the information regarding the amount of GST-GH3.5 to the Methods section.

3. The kinetics data should be conducted in triplicate, which is unclear from the singlet points and absence of error bars on the plot in Figure 5f.

Response:

Thank you for the comment. In fact, the kinetic analysis shown in Fig. 5f was performed in triplicate, with only one representative result shown in the figure, as indicated in the figure legends. The results from all three experiments are now shown in the revised Fig. 5f.

Fig. 5f Michaelis–Menten diagram and kinetic parameters of GH3.5 synthetase for IAA-Asp. Initial velocities are plotted against the IAA concentration. The average values \pm SD ($n = 3$) are shown.

4. GH3.5 is promiscuous and can conjugate amino acids onto multiple auxins (IAA and phenylacetic acid) and benzoates (salicylic acid and benzoic acid) in vitro and in vivo. In this publication (www.pnas.org/cgi/doi/10.1073/pnas.1612635113), GH3.5 has a higher catalytic efficiency with BA relative to IAA. Perhaps it would be worth measuring BA-Asp, PAA-Asp, and SA-Asp to determine whether other products of GH3.5 are affected by light and dark.

Response:

Thank you for your comment. We have carefully reviewed the data from the cited publication (www.pnas.org/cgi/doi/10.1073/pnas.1612635113). GH3.5 catalyzed the conjugation of SA ($k_{cat}/K_m = 19 \text{ M}^{-1} \text{ s}^{-1}$), BA ($k_{cat}/K_m = 338 \text{ M}^{-1} \text{ s}^{-1}$), and IAA ($k_{cat}/K_m =$

314 M⁻¹ s⁻¹), suggesting that GH3.5 exhibited comparable catalytic efficiency with BA and IAA, but substantially lower catalytic efficiency with SA. IAA is the predominant natural auxin responsible for promoting cell expansion in plants¹. Our genetic and biochemical analyses show that light modulates IAA metabolism by suppressing COP1-mediated inhibition of GH3.5 enzyme activity. It would be interesting to investigate in the future whether light signaling also regulates the catalytic activity of GH3.5 with other acyl acid substrates.

The references mentioned in the response are listed below:

1. Casanova-Saez R, Mateo-Bonmati E, Ljung K. Auxin metabolism in plants. *Cold Spring Harbor Perspect Biol* **13**, (2021).

5. In Figure 5a, the 0 mM data should also be included for comparison.

Response:

Thank you for the comment. The original data for Fig. 5a are shown in Fig. R4 below. Seedlings grown in the dark were treated with 0 μM, 0.1 μM, or 0.2 μM IAA. In Fig. 5a, IAA-mediated inhibition of root growth was expressed as the percentage decrease in the root length of IAA-treated seedlings compared to control seedlings. We have revised the figure legend for Fig. 5a to clarify this.

Fig. R4 Auxin inhibition of primary root growth in Col, *cop1-4*, *gh3.5-1*, and *cop1-4 gh3.5-1*. Seedlings were grown in the dark on MS medium supplemented with or without IAA. The IAA concentration varied as follows: 0, 0.1, and 0.2 μM .

6. More information should be provided in the methods regarding protein expression, purification, the buffers that were used, and how protein concentration was determined.

Response:

Thank you for your suggestion. Detailed information on prokaryotic expression and purification of proteins has been added to the Methods section.

Minor point:

1. On lines 255 and 258: “catalytic rate” is used to describe the kinetics, which will need to be modified unless steady-state, continuous assays are used instead of endpoint assays.

Response:

Thank you for your comment. We have replaced “catalytic rate” with “catalytic activity” in the revised manuscript.